# Breaking the Data Barrier – Building GUI Agents Through Task Generalization

**Junlei Zhang**[*◇☆] **Zichen Ding**[*♠] **Chang Ma**[♣] **Zijie Chen**[◇☆] **Qiushi Sun**[♣]
**Zhenzhong Lan**[☆] **Junxian He**[★]
[◇]Zhejiang University  [☆]Westlake University  [♠]Shanghai AI Laboratory
[♣]The University of Hong Kong  [★]HKUST

## Abstract

Graphical User Interface (GUI) agents offer cross-platform solutions for automating complex digital tasks, with significant potential to transform productivity workflows. However, their performance is often constrained by the scarcity of high-quality trajectory data. To address this limitation, we propose training Vision Language Models (VLMs) on data-rich, reasoning-intensive tasks during a dedicated mid-training stage, and then examine how incorporating these tasks in the mid-training phase facilitates generalization to GUI planning scenarios. Specifically, we explore a range of tasks with readily available instruction-tuning data, including GUI perception, multimodal reasoning, and textual reasoning. Through extensive experiments across 11 mid-training tasks, we demonstrate that: (1) Task generalization proves highly effective, yielding substantial improvements across most settings. For instance, multimodal mathematical reasoning enhances performance on AndroidWorld by an absolute 6.3%. Remarkably, text-only mathematical data significantly boosts GUI web agent performance, achieving a 5.6% improvement on WebArena and a 5.4% improvement on AndroidWorld, underscoring notable cross-modal generalization from text-based to visual domains; (2) Contrary to prior assumptions, GUI perception data—previously considered closely aligned with GUI agent tasks and widely utilized for training—has a comparatively limited impact on final performance; (3) Building on these insights, we identify the most effective mid-training tasks and curate optimized mixture datasets, resulting in absolute performance gains of 8.0% on WebArena and 12.2% on AndroidWorld. Our work provides valuable insights into cross-domain knowledge transfer for GUI agents and offers a practical approach to addressing data scarcity challenges in this emerging field. The code, data, and models are available at https://github.com/hkust-nlp/GUIMid.

## 1 Introduction

Interacting with graphical user interfaces (GUIs) has become a fundamental part of how humans engage with the world, from browsing the internet to using mobile apps. Developing autonomous agents capable of seamlessly interacting with these interfaces as personal assistants has the potential to revolutionize daily life (Xie et al., 2024; OpenAI, 2025; Wu et al., 2024), making it more efficient and convenient.

Building GUI agents requires a combination of key capabilities: perception—understanding and interpreting GUI screenshots, grounding—translating human instructions into executable actions, and visual planning—carrying out tasks step by step to achieve the desired goal (Zheng et al., 2024b; Xu et al., 2024b; Ma et al., 2024). Among these, visual planning is the most challenging (Gur et al., 2023; Koh et al., 2024b; Yu et al., 2024). It demands breaking down complex instructions, like "check my GitHub repositories with the most stars," into

---

[*]Co-first author. Work done during JZ's visit to HKUST. Correspondence to Junlei Zhang (zhangjunlei@westlake.edu.cn) and Junxian He (junxianh@cse.ust.hk).

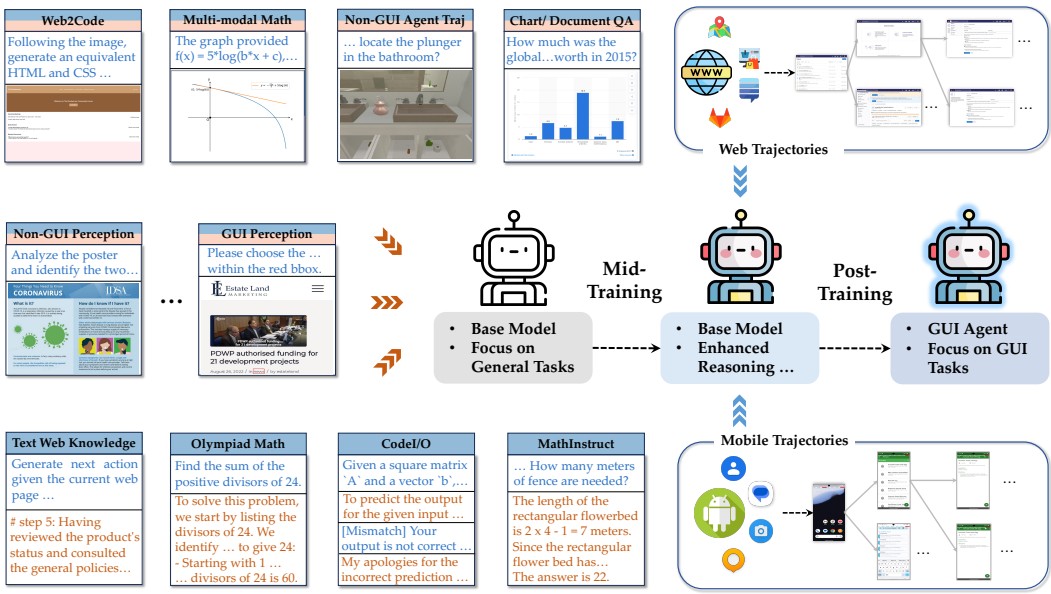

Figure 1: Overview of the mid-training and fine-tuning process. Left: We first train the GUI agent on mid-training data, primarily from non-GUI domains, to investigate whether the enhanced capabilities can generalize to GUI agent tasks; Right: We perform post-training on GUI trajectory data.

precise, actionable steps. Vision-Language models (VLMs) naturally have the potential to serve as policy models for guiding GUI agent planning. With advanced prompting techniques, they can act as the foundational models for GUI agents (Zheng et al., 2024b; Song et al., 2024). However, most existing VLMs lack the reliability and stability needed to perform as effective GUI agents, often delivering subpar performance on benchmarks (Zhou et al., 2023; Deng et al., 2023; Koh et al., 2024a; Xie et al., 2024). For example, gpt4-o (Hurst et al., 2024) achieves a mere 15.6% on WebArena (Zhou et al., 2023), and a Qwen2-VL-7B-Instruct (Wang et al., 2024a) could barely generate goal-aligned actions. Most errors stem from insufficient planning when tasks require multiple sequential steps. To address this limitation, researchers have been trying to improve the GUI policy VLMs by collecting or synthesizing GUI trajectory data (Xu et al., 2024b;a; Sun et al., 2024b; Ou et al., 2024). However, real-world GUI trajectory data is not readily available, making the acquisition of diverse and high-quality GUI datasets a significant challenge. Additionally, synthesizing trajectories using large models often results in low-quality outputs, as even the most advanced VLMs struggle to perform effectively on realistic GUI agent tasks. In light of this challenge, we focus our study on the critical question: **How to improve the agentic abilities of VLMs for GUI tasks with more scalable data sources?**

To this end, we introduce a **mid-training** stage to enhance the foundational agentic capabilities of VLMs prior to fine-tuning them on a limited subset of GUI trajectories for task-specific adaptation. Mid-training (illustrated in Figure 1) refers to an intermediate training stage between pre-training and fine-tuning, where models are enhanced with specialized capabilities for better adaptation.[1] While this approach has been successfully applied in previous studies across various scenarios involving LLMs and VLMs (Wang et al., 2024b; Sun et al., 2024a; Yang et al., 2025), it remains unclear which types of tasks can be effectively generalized to learning GUI agents, given the complexity and typically extensive context associated with such tasks. Previous research exploring alternative data sources has primarily focused on GUI-specific resources, such as web tutorials and GUI captions (Chen et al., 2024; Xu et al., 2024a; Ou et al., 2024), which falls short on representing

---

[1]The definition of mid-training can differ, conceptually it is similar to continual pretraining on instruction tuning data in our context.

agentic planning abilities. In this work, we investigate a diverse set of mid-training data domains to evaluate their impact on learning GUI agents. These domains include general image perception, chart understanding, multimodal reasoning, and text-based tasks such as mathematics and programming.

We evaluate eleven datasets—seven multimodal and four textual—focusing on reasoning, knowledge retrieval, and perception (§3.1). Samples are collected per domain, followed by separate mid-training on each, and fine-tuning on a GUI trajectory dataset. By standardizing all datasets to generate high-level action thoughts before grounding to actions, we enhance planning transfer. A continuous optimizer ensures smooth training transitions and minimizes forgetting. Analysis on mobile and web benchmarks validates the effectiveness of our mid-training approach. Pure text mathematical datasets show the largest gains, improving AndroidWorld by 5.4% and WebArena by 5.6%, demonstrating reasoning abilities could transfer cross-domain. Coding datasets boost performance by around 3.0% on both tasks. Surprisingly, visual perception datasets yield modest gains, likely due to existing VLMs' strong visual capabilities. Based on these insights, we introduce GUIMid, a 300k dataset combining the four best-performing domains. GUIMid achieves SOTA on AndroidWorld in pure-visual settings and improves Qwen2-VL to GPT4-o level performances on web browsing, with overall gains of 12.2% and 8.0% on AndroidWorld and WebArena.

## 2 Vision-based GUI Agent Framework

**Pure vision-based GUI agents.** Previous work (Gur et al., 2023; Zheng et al., 2024b; Xie et al., 2024; Zhou et al., 2023) has largely relied on structural text-based GUI representations, such as HTML or accessibility trees. In contrast, we focus on the more challenging pure-vision setting, where vision-based agents take screenshots and task descriptions as input, generating coordinate-based actions directly within pixel space. This pure-vision approach provides key advantages: (1) it eliminates dependencies on backend structures, enabling cross-platform operation while avoiding the noise often present in accessibility trees, and (2) it aligns more closely with human interaction patterns, allowing for seamless integration into real-world workflows.

The inference process can be formalized as a specialized instance of Partially Observable Markov Decision Processes (POMDPs), represented by tuple $\langle g, \mathcal{S}, \mathcal{A}, \mathcal{O}, \mathcal{T} \rangle$, where $g$ denotes the task goal, $\mathcal{S}$ the state space, $\mathcal{A}$ the action space, $\mathcal{O}$ the observation space (visual feedback from the screen), and $\mathcal{T} : \mathcal{S} \times \mathcal{A} \rightarrow \mathcal{S}$ the state transition function. At each time step $t$, the agent executes decisions according to policy $\pi$, which integrates the task goal $g$, memory $m_t = \{o_j, a_j, o_{j+1}, a_{j+1}, \ldots, o_{t-1}, a_{t-1}\}, 0 \leq j < t$ (capturing the history of actions and observations), and the current observation $o_t$. The agent's trajectory, denoted as $\tau = [s_0, a_0, s_1, a_1, \ldots, s_t]$, emerges from the policy and environmental state transitions, as formulated by:

$$p_\pi(\tau) = p(s_0) \prod_{t=0}^{T} \pi(a_t \mid g, s_t, m_t) \mathcal{T}(s_{t+1} \mid s_t, a_t) \tag{1}$$

We then introduce specific implementations of the observation and action space in our vision-based GUI agent framework.

**The action space.** Our GUI agent employs a coordinate-based action space to ensure cross-platform compatibility and realistic human-like interactions. The policy model generating actions consists of two components: a planner model and a grounding model. The planner model first generates high-level action description following a planning-rich thought, i.e. "Task instruction <thought> <high-level action>". Then the grounding model maps the high-level action into screenshot manipulations.

For instance, to search for "GUI agents" in a search engine, the generated content is:

```
<thought>: To find unlabeled issues in the metaseq GitLab repository, click the
"Issues" tab in the main navigation menu, then filter for issues without labels.
```

| Domains | Ability | Datasets | Samples | Type |
|---|---|---|---|---|
| **Vision-and-Language Modality** | | | | |
| **Chart/Document QA** | Perception | InfographicVQA (Guo et al., 2024) | 2,184 | Instruction, Thought[*], Answer |
| | | Ureader QA (Guo et al., 2024) | 53,794 | Instruction, Thought, Answer |
| | | MPDocVQA (Tito et al., 2023) | 431 | Instruction, Thought, Answer |
| | | MathV360k (Liu et al., 2024b) | 93,591 | Instruction, Thought, Answer |
| **Non-GUI Perception** | Perception | Ureader OCR (Ye et al., 2023) | 6,146 | Instruction, Thought[*], Answer |
| | | DUE (Borchmann et al., 2021) | 143,854 | Instruction, Answer |
| **GUI Perception** | Perception | MultiUI (Liu et al., 2024a) | 150,000 | Instruction, Answer |
| **Web Screenshot2Code** | Perception | Web2Code (Yun et al., 2024) | 150,000 | Instruction, Answer |
| **Multi-modal Math** | Reasoning | Mavis (Zhang et al., 2024b) | 150,000 | Instruction, Thought, Answer |
| **Multi-round Visual Conversation** | Interaction | SVIT (Zhao et al., 2023) | 150,000 | Instruction, Thought, Answer |
| **Non-GUI Agent Trajectories** | Interaction | AlfWorld (Guo et al., 2024) | 51,780 | Instruction, Thought, Answer |
| **Language Modality** | | | | |
| **MathInstruct** | Reasoning | MathInstruct (Yue et al., 2023) | 150,000 | Instruction, Thought, Answer |
| **Olympiad Math** | Reasoning | NuminaMath (LI et al., 2024) | 150,000 | Instruction, Thought, Answer |
| **CodeI/O** | Reasoning | CodeI/O (Li et al., 2025) | 150,000 | Instruction, Thought, Answer |
| **Web Knowledge Base** | Knowledge | Synatra (Ou et al., 2024) | 99,924 | Instruction, Thought, Answer |
| | | AgentTrek (Xu et al., 2024a) | 50,076 | Instruction, Thought, Answer |

Table 1: Statistics of the domains and corresponding datasets used in mid-training, (*) indicates that some instructions in the dataset do not require a "Thought" (e.g., "Answer concisely with one word or phrase.").

```
<high-level action>:
{
    "Element Description": "Click the Issues tab in the main navigation menu",
    "Action": "click",
}

<grounded action>: Click [coordinate_x 0.12]  [coordinate_y 0.07]
```

We use UGround-V1-7B (Gou et al., 2025) for grounding, and our trained policy model for thought and high-level action generation. This separation also enables better transfer of planning ability through mid-training (as detailed in §3) in addition to flexible inclusion of new actions. The details of action spaces for mobile and web tasks can be found in Table 9, 10.

**The observation space and memory.** The observation space is visual-rich in our framework, comprising of a screenshot of the current screen and simple meta-information (i.e., the current URL for web tasks). To augment agent memory, we also provide the model with a history of previous actions, using the concatenated output of planner generated high-level actions, e.g. "step 1: click 'the search results titled with wikipedia'; step 2: type 'GUI Agent' into the search bar at the top of the page".

## 3 Breaking the Data Barrier via Mid-Training

Despite advancements in vision-language models, GUI agent training faces challenges due to limited high-quality trajectory data. We introduce a mid-training stage between general pre-training and task-specific post-training. This approach leverages abundant data from adjacent domains—image perception, chart understanding, multimodal reasoning, and programming—to develop foundational capabilities before GUI-specific adaptation. Our two-step strategy includes mid-training on scalable data sources (§3.1) followed by fine-tuning on a small GUI trajectory dataset (§3.2). Our optimized training procedure introduced in §3.3 ensures consistent generalization to GUI agent skills. We briefly introduce these datasets and the training procedure in this section, and more details can be found in Appendix B, C.

| Domains | Datasets | Samples | Type |
|---------|----------|---------|------|
| **Web** | OS-Genesis (Web) (Sun et al., 2024b) | 3,789 | Instruction, Thought, Action |
| | MM-Mind2Web (Zheng et al., 2024a) | 21,542 | Instruction, Thought, Action |
| | VisualWebArena (Koh et al., 2024a) | 3,264 | Instruction, Thought, Action |
| **Mobile** | OS-Genesis (Mobile) (Sun et al., 2024b) | 4,941 | Instruction, Thought, Action |
| | Aguvis (Xu et al., 2024b) | 22,526 | Instruction, Thought, Action |

Table 2: Statistics of the web/mobile domains along with the corresponding GUI trajectory datasets used in post-training.

### 3.1 Mid-Training Data

We collect 150k diverse training samples for each domain to study cross-domain generalization. For the non-GUI agent domain, we included 51k samples due to the scarcity of agent trajectories. The mid-training domains are listed in Table 1. For vision-language tasks, we include: Chart/Document QA (Guo et al., 2024; Tito et al., 2023) and Multi-modal Math (Zhang et al., 2024b) for fine-grained understanding and visual reasoning; Non-GUI Perception tasks (Ye et al., 2023; Borchmann et al., 2021) including Document OCR for fundamental visual understanding; Web Screenshot2Code (Yun et al., 2024) for structured web screenshot interpretation; and multi-turn data from Visual Conversations (Zhao et al., 2023) and Non-GUI Agent Trajectories (Guo et al., 2024) to enhance VLM interactive capabilities.

Complementing these, we include pure-text data featuring more reasoning-intensive, and knowledge-rich tasks, including from medium-difficulty (MathInstruct (Yue et al., 2023)) to challenging (Olympiad Math (LI et al., 2024)) mathematical reasoning, Code I/O (Li et al., 2025) to develop procedural reasoning through code generation, and Web Knowledge Base (Ou et al., 2024; Xu et al., 2024a) to inject domain knowledge.

### 3.2 Post-Training GUI Trajectory Data

For post-training, we used high-quality GUI trajectories from state-of-the-art systems across platforms. We incorporated web and mobile data from OS-Genesis (Sun et al., 2024b), Aguvis (Xu et al., 2024b), and MM-Mind2Web (Zheng et al., 2024a), plus 3.2K manually annotated steps from VisualWebArena (Koh et al., 2024a). Our final dataset contains 28K web samples and 27K mobile samples, as detailed in Table 2.

### 3.3 Training Method

We employ a two-stage training approach consisting of mid-training followed by fine-tuning on GUI trajectory data. Both stages are integrated under a single optimizer and learning rate schedule. During the mid-training stage, we mix the post-training GUI trajectory data into the mid-training domain data (e.g. ChartQA) to mitigate potential forgetting issues, which we will analyze empirically in §4.4. This mixing is particularly crucial when there exists a substantial domain gap between the mid-training data and GUI trajectory data, especially when the former is primarily text-based (see Figure 4 for comparison). Additional training hyperparameters and details are detailed in Appendix D.

## 4 Experiments

### 4.1 Experimental Setup

To explore how non-GUI trajectory data can enhance VLMs' foundational agentic capabilities, we study 7 multi-modal domains and 4 language domains (Table 1). For the post-training data, we collect 56K high-quality data (Table 2). Following State-Of-The-Art (SOTA) works (Xu et al., 2024b; Sun et al., 2024b; Gou et al., 2025), we employ Qwen2-VL-7B-Instruct (Wang et al., 2024a) as our backbone. We first train our model on the mid-training datasets, then fine-tune it on GUI trajectories. To study whether these separate domains can

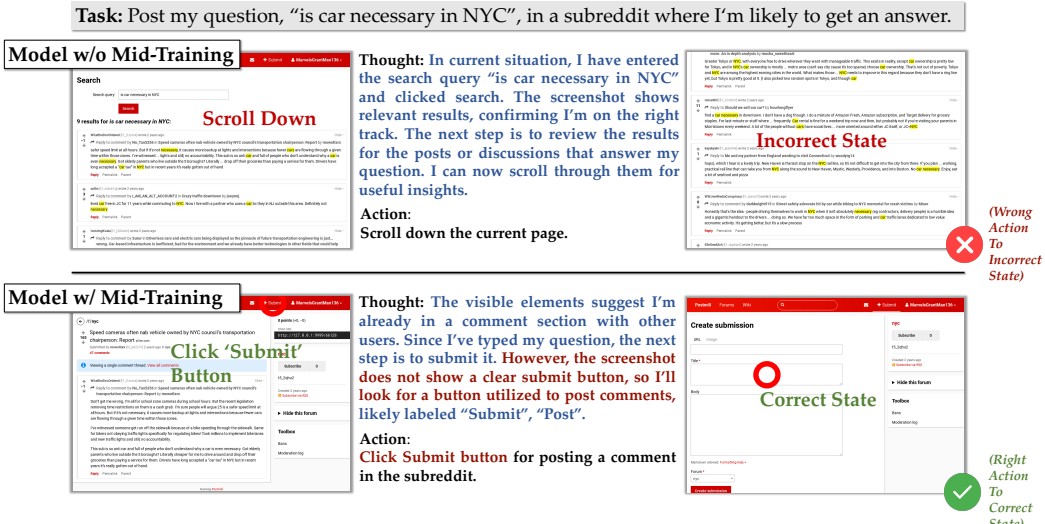

Figure 2: A case illustrating the performance of the Model w/o Mid-Training and the Model w/ Mid-Training under the same task. The middle text shows the model's thought process and the action taken, while the screenshots on the left and right represent the screen states before and after the action, respectively. The model with middle training (bottom) successfully reflects on errors and generates correct actions from error states, while the model without mid-training (top) fails to recover from such states.

be combined to achieve superior performance, we randomly sample a total of 300K examples from high-performing domains (150K from MathInstruct, 20K from CodeI/O, 50K from Olympiads Math, and 80K from Multi-modal Math) to create a consolidated mid-training dataset called **GUIMid**. During the mid-training stage, we mix the GUI trajectory samples into our mid-training dataset to make the training stable.

## 4.2 Evaluation Environment

We use AndroidWorld (Rawles et al., 2024) and WebArena (Zhou et al., 2023) as our testbeds, as their dynamic nature makes them ideal environments for study the effects of different domains during the mid-training stage. For WebArena, we use the version processed in AgentBoard (Ma et al., 2024). We opt for interactive benchmarks over static benchmarks (Deng et al., 2023; Li et al., 2024) based on two considerations: (1) Static environments inherently contain annotation bias, wherein multiple valid paths exist to complete a task, yet annotator preferences result in only one path being labeled as correct for each step (e.g., finding a playlist through either search functionality or category navigation); (2) Our mid-training approach primarily enhances fundamental capabilities such as reasoning and perception rather than in-domain knowledge—for example, improving the model's ability to complete tasks in new websites through exploration and reflection. In contrast, step-by-step optimal action annotations rely heavily on the model's prior knowledge about specific websites or applications. We provide more evaluation details are provided in Appendix E.

## 4.3 Results on Separate Domain

We report the performances of different domains as mid-training on Webarena and Android-World (Table 3). Our analysis reveals several important findings:

**Mathematical data generally improved the most:** Both language-only and vision-language mathematical reasoning tasks demonstrate substantial performance improvements across benchmarks. In the language modality, models mid-trained with MathInstruct achieve the highest success rate on WebArena (10.9%) and strong performance on AndroidWorld (14.4%). Similarly, in the vision-language domain, Multi-modal Math shows

| Domains | Observation | WebArena | | AndroidWorld |
|---|---|---|---|---|
| | | PR | SR | SR |
| **GUI Post-Training Only** | Image | 26.3 | 6.2 | 9.0 |
| **Public Baselines** | | | | |
| **GPT-4o-2024-11-20** | Image | 36.9 | 15.6 | 11.7 |
| **OS-Genesis-7B** | Image + Accessibility Tree | – | – | 17.4 |
| **AGUVIS-72B** | Image | - | - | 26.1 |
| **Claude3-Haiku** | Accessibility Tree | 26.8 | 12.7 | - |
| **Llama3-70b** | Accessibility Tree | 35.6 | 12.6 | - |
| **Gemini1.5-Flash** | Accessibility Tree | 32.4 | 11.1 | - |
| **Vision-and-Language Modality** | | | | |
| **Chart/ Document QA** | Image | 24.6 | 6.2 | 15.3 |
| **Non-GUI Perception** | Image | 28.7 | 7.6 | 14.0 |
| **GUI Perception** | Image | 27.4 | 7.1 | 14.0 |
| **Web Screenshot2Code** | Image | 28.0 | 6.6 | 9.9 |
| **Non-GUI Agents** | Image | 30.8 | 8.5 | 13.5 |
| **Multi-modal Math ✓** | Image | 30.4 | 8.5 | 15.3 |
| **Multi-round Visual Conversation** | Image | 30.0 | 9.0 | 12.6 |
| **Language Modality** | | | | |
| **MathInstruct ✓** | Image | 31.9 | 10.9 | 14.4 |
| **Olympiad Math ✓** | Image | 31.5 | 8.5 | 13.1 |
| **CodeI/O ✓** | Image | 29.2 | 9.0 | 14.9 |
| **Web Knowledge Base** | Image | 31.3 | 9.5 | 9.0 |
| **Domain Combination (Sampled data from ✓ domains)** | | | | |
| **GUIMid** | Image | 34.3 | 9.5 | 21.2 |

Table 3: Progress Rate (PR) and Success Rate (SR) of *Qwen2-VL-7B-Instruct* across various domains using a two-stage training strategy. Color-coded cells (green/red) are employed to denote improvements or declines relative to the post-training only baseline, with deeper shades indicating larger score shifts.

impressive gains of 8.5% on WebArena and 15.3% on AndroidWorld. This consistent pattern suggests that mathematical reasoning capabilities, regardless of input modality, improve generalizable reasoning skills that transfer effectively to GUI agent tasks. A case study of a model mid-trained by MathInstruct (Yue et al., 2023) is shown in Figure 2. The task asks the agent to post the question "Is car necessary in NYC" on the Reddit website. Both agents initially navigated to incorrect pages. However, the MathInstruct-trained agent demonstrated better reasoning by thinking, "However, the screenshot does not show a clear submit button, so I'll look for a button utilized to post comments," and subsequently located the correct "submit" button. In contrast, the baseline model without mid-training became stuck on the incorrect page and continued scrolling up and down fruitlessly.

**Strong cross-modal and cross-domain transfer:** Language-only tasks demonstrate remarkable effectiveness for multi-modal GUI tasks. Olympiad Math achieves 31.5% progress and 8.5% success on WebArena, outperforming most vision-language tasks. Similarly, CodeI/O reaches a 14.9% success rate on AndroidWorld. Web Knowledge Base shows effectiveness primarily in the web domain, likely due to its focus on web-specific information rather than mobile. These results suggest that conduct mid-training on text-only mathematics, code, and knowledge data can enhance fundamental abilities for GUI agents, even for multi-modal tasks—offering valuable insights for addressing the challenge of limited GUI in-domain training data.

Non-GUI Agents data, despite its relatively small size ( 50K samples), demonstrates strong performance (30.8% progress, 8.5% success on WebArena; 13.5% success on AndroidWorld). This suggests that agent interaction patterns can transfer to some extent. Multi-round Visual Conversation yields balanced improvements across benchmarks (9.0% on WebArena, 12.6% on AndroidWorld). Chart/Document QA performs good on AndroidWorld(15.3%) but bad on the web setting ,suggesting the different requirement of digital platforms. However, the Web Screenshot2Code and GUI Perception data does not help a lot, this may due to Qwen2-VL-7B-Instruct already be trained on GUI perception data.

**Results on Domain Combination** Our motivations is to leverage larger quantities of non-GUI datasets to enhance VLMs' foundational capabilities, thereby enabling more

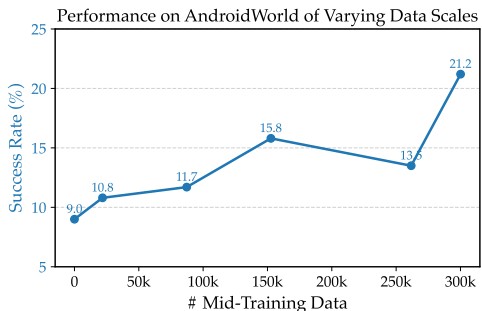 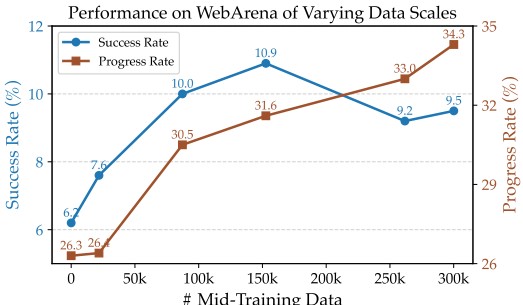

Figure 3: Performance of models trained on GUIMidwith different scales.

| Training Strategy | Success Rate (%) |
|---|---|
| **GUI Post-Training Only** | 6.1 |
| **GUI Mid + Post-Training** | 10.2 |

Table 4: Performance Comparison on Mind2Web-Live

| Model | Allrecipes | Amazon | Apple | ArXiv |
|---|---|---|---|---|
| GUI Post-Training Only | 30.0 | 10.0 | 20.0 | 50.0 |
| GUI Mid + Post-Training | **50.0** | **40.0** | **40.0** | **50.0** |
| | **BBC News** | **Booking** | **Cambridge Dictionary** | **Coursera** |
| GUI Post-Training Only | 10.0 | 10.0 | 20.0 | 50.0 |
| GUI Mid + Post-Training | **20.0** | **20.0** | **70.0** | **50.0** |
| | **ESPN** | **GitHub** | **Google Flights** | **Google Map** |
| GUI Post-Training Only | 10.0 | 20.0 | 0.0 | 10.0 |
| GUI Mid + Post-Training | **20.0** | **40.0** | **10.0** | **20.0** |
| | **HuggingFace** | **Wolfram Alpha** | **Average** | |
| GUI Post-Training Only | 30.0 | 20.0 | 20.7 | |
| GUI Mid + Post-Training | **50.0** | **50.0** | **37.9** | |

Table 5: Performance Comparison on WebVoyager.

effective learning from limited GUI trajectory data. To validate this, we combine the top-performing domains to construct a combined mid-training dataset: **GUIMid**, which consists of randomly sampled data (150K samples from MathInstruct, 20K from CodeI/O, 50K from Olympiads Math, and 80K from Multi-modal Math). We explore the scaling law of GUIMid. Specifically, for the 300K sample mid-training dataset, we maintain the same ratio of non-GUI to GUI data as in our 150K sample experiments to ensure training stability. Instead of introducing new GUI data, we duplicate the existing GUI trajectory data. In Figure 3, the x-axis represents the effective mid-training data volume, calculated as the total training data volume multiplied by the fixed proportion allocated to mid-training samples. The model exhibits scaling laws with increasing mid-training data volume on both AndroidWorld and WebArena. While we observe a slight decrease in WebArena success rates around the 300K sample mark, the progress rate metric provides a more nuanced assessment of performance improvements. This metric, which captures fine-grained capability development, shows consistent growth: from approximately 26.4% at 21.8K samples to 31.6% at 152K samples, and further to 34.3% at 300K samples. This steady upward trajectory indicates the effective of the scaling law of GUIMid.

To further validate our method, we evaluated it on Mind2Web-Live (Pan et al., 2024) and WebVoyager (He et al., 2024) without additional fine-tuning. On Mind2Web-Live, we ran 51 accessible tasks, excluding those restricted by website access, and—because ground-truth

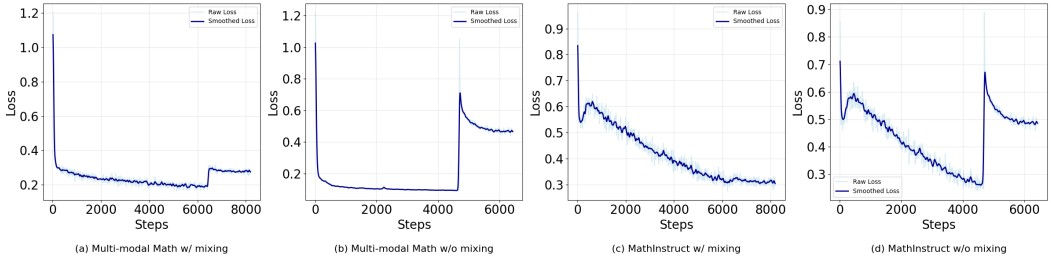

Figure 4: Comparison of training loss between two training strategies: (a) and (c) show the mixture of GUI trajectory data during mid-training, while (b) and (d) are not.

| Domains | WebArena | | AndroidWorld |
|---|---|---|---|
| | PR | SR | SR |
| **MathInstruct (no mixing)** | 24.0 | 9.0 | 9.0 |
| **MathInstruct (mixing)** | 33.6 | 8.5 | 14.4 |
| **Multi-modal Math (no mixing)** | 25.4 | 6.2 | 14.9 |
| **Multi-modal Math (mixing)** | 30.4 | 8.5 | 15.3 |

Table 6: Progress Rate (PR) and Success Rate (SR) with and without GUI trajectory data integration during the mid-training stage. "mixing" indicates the mid-training data is mixed with GUI trajectory data, while "no mixing" indicates it was not.

| Domains | Difficulty | WebArena | | AndroidWorld |
|---|---|---|---|---|
| | | PR | SR | SR |
| **Orca-Math** | Easy | 31.9 | 10.0 | 9.9 |
| **Randomly Sampled Data** | Middle | 30.6 | 9.5 | 10.8 |
| **Olympiad Math** | Hard | 31.5 | 8.5 | 13.1 |

Table 7: The impact of mathematical difficulty in mid-training data. We sample three subsets from the NuminaMath dataset based on their difficulty levels.

labels can become obsolete as websites evolve—we report success rate as the primary metric. On WebVoyager, we randomly sampled ten tasks from each of fourteen domains (140 tasks in total), omitting Google Search due to reCAPTCHA limitations, and followed the benchmark's standard human-evaluation protocol. Across both benchmarks our method delivers substantial gains: on Mind2Web-Live it achieves a 67 % relative improvement (10.2 vs 6.1), while on WebVoyager it yields an 83 % average relative improvement (37.9 vs 20.7), with particularly large gains in domains that demand complex reasoning, such as Cambridge Dictionary (+250 %) and Wolfram Alpha (+150 %).

### 4.4 Ablation Study

**The effects of adding GUI trajectory data to the mid-training stage.** We compare the loss curves (Figure 4) and performances (Table 6) when mixing/no mixing GUI trajectory data into the mid-training stage. Due to the domain gap between mid-training data and GUI trajectory data, domain switching may cause sharp fluctuations in the loss curve, leading to instability (as shown in Figure 4, plots (b) and (d)), or even cause gradient overflow to NaN values. In contrast, plots (a) and (c) demonstrate significantly smoother training dynamics after merging the data. Table 6 reveals that after mixing GUI trajectory data, both MathInstruct and Multi-modal Math show significant performance improvements. In WebArena, MathInstruct's progress rate increased from 24.0 to 33.6 after mixing. Also, in AndroidWorld, Multi-modal Math's success rate improved from 14.9 to 15.3 after mixing.

**Performance with Respect to Mid-training Data Difficulty** We analyze performance across three NuminaMath subsets of varying difficulty: the relatively easier Orca-Math (Mitra et al., 2024), the highly challenging Olympiad Math (LI et al., 2024), and a medium-

| Training Strategy | Post-training Data Ratio | WebArena (Progress Rate) | WebArena (Success Rate) | AndroidWorld (Success Rate) |
|---|---|---|---|---|
| GUI Post-Training Only | 100% | 26.3 | 6.2 | 9.0 |
| GUI Mid + Post-training | 20% | 25.5 | 6.6 | 10.8 |
| GUI Mid + Post-training | 60% | 27.0 | 8.1 | 12.6 |

Table 8: Comparison of post-training data efficiency.

difficulty random subset. Table 7 shows that performance in mobile environments generally improves with increased data difficulty, while WebArena exhibits no clear correlation. This difference likely stems from AndroidWorld's more diverse interaction patterns.

**Post-training efficiency comparison with and without mid-training.** We controlled the amount of in-domain trajectory data used in both the mid-training and post-training stages by randomly sampling different proportions of the original post-training dataset. Table 8 shows that our *GUI Mid + Post-training* approach achieves performance comparable to the baseline while using only 20 % of the post-training data (WebArena: progress rate 25.5 vs 26.3, success rate 6.6 vs 6.2; AndroidWorld: 10.8 vs 9.0).

# 5 Related Work

**GUI Agents.** Recent advancements in GUI agents have spurred the development of diverse benchmarks. Early non-interactive benchmarks for web (Mind2web (Deng et al., 2023), WebLINX (Lu et al., 2024b), WebVoyager (He et al., 2024)) and mobile (AITW (Rawles et al., 2023), AITZ (Zhang et al., 2024a), AndroidControl (Li et al., 2024)) suffer from potential annotator bias in tasks with multiple valid solutions. To mitigate this, interactive benchmarks were introduced for desktop and mobile (OSWorld (Xie et al., 2024), AndroidWorld (Rawles et al., 2024)) and web environments (WebArena (Zhou et al., 2023), VisualWebArena (Koh et al., 2024a), WorkArena (Drouin et al., 2024), WebCanvas (Pan et al., 2024)). However, these frameworks pose challenges for evaluating specific non-GUI capabilities like reasoning. For instance, VisualWebArena's trajectory-level success metrics can mask reasoning improvements when perception is the primary bottleneck. While WebCanvas (Pan et al., 2024) aims to address perceptual challenges, many of its tasks are outdated due to live site updates. Therefore, to isolate and measure specific capabilities such as reasoning, we adopt the subgoal-annotated versions of WebArena and AndroidWorld from AgentBoard (Ma et al., 2024), which enables a more granular evaluation independent of perceptual limitations.

**Mid-Training.** Mid-training refers to the stage between pre-training and post-training designed to enhance foundational capabilities (Abdin et al., 2024), whereas post-training optimizes models to adapt to specific downstream tasks. Recent works like *Phi 3.5* (Abdin et al., 2024), *Yi-Lightning* (Wake et al., 2024), *OLMo 2* (OLMo et al., 2024) and CodeI/O (Li et al., 2025) have demonstrated the effectiveness of strategic mid-training interventions-use mid-training to enhance the foundational abilities like context length, multilingual knowledge and code understanding. For GUI agents, where high-quality trajectories are scarce, mid-training becomes particularly valuable. However, current research lacks systematic exploration of out-of-domain mid-training techniques specifically tailored for GUI agents—a critical gap that our work addresses.

# 6 Conclusion

In this paper, we propose using mid-training to enhance GUI agents' foundational capabilities, enabling more effective learning from limited trajectory data. While GUI-specific data remains scarce, there is much more non-GUI data such as mathematical reasoning, and coding. We explore 11 diverse non-GUI tasks, demonstrating for the first time the significant impact of mathematical reasoning data on GUI task performance, as well as the surprising effectiveness of text-only mathematical reasoning in improving multimodal GUI agent capabilities. Our findings also reveal how non-GUI tasks such as text web knowledge and embodied agent trajectories can substantially enhance GUI agent performance. These results provide valuable insights for the research community in constructing more effective GUI training pipelines. We will open-source all data, models, and training recipes to facilitate future research in the GUI agent domain.

## 7 Acknowledgements

This project is supported by NSFC Grant 62306177 and the program "Development of Digital Therapeutic Systems Based on Large Language Models and Their Application to Major Brain Diseases" (Project No. 2025C01104).

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

| Action | Description |
|---|---|
| click [[x] [y]] | Click at coordinates (x, y). |
| type [[x] [y]] [value] | Type content into a field by coordinate. |
| scroll [[x] [y]] [value] | Scroll the page or a specific element. |
| go_back | Navigate to the previous screen or page. |
| go_home | Navigate to the home screen. |
| long_press[[x] [y]] | Long press on an coordinate. |
| enter | Press the Enter key. |
| open_app [app_name] | Open an app by [app_name]. |
| wait [value] | Wait for the screen to update for [value] seconds. |
| stop [answer] | Stop the task with a goal status or answer. |

Table 9: The action space for mobile tasks.

| Action | Description |
|---|---|
| click [[x] [y]] | Click at coordinates (x, y). |
| type [[x] [y]] [value] | Type content into a field by coordinate. |
| clear [[x] [y]] | Clear the content of an element. |
| hover [[x] [y]] | Hover over an element by coordinate. |
| press [keys] | Press a key combination (e.g., Ctrl+v). |
| scroll [value] | Scroll the page. |
| new_tab | Open a new tab. |
| page_focus [tab_index] | Switch to a specific tab. |
| close_tab | Close the current tab. |
| goto [url] | Navigate to a URL. |
| go_back | Go to the previous page. |
| go_forward | Go to the next page. |
| stop [answer] | Issue this action when the task is considered complete. |

Table 10: The action space for web tasks.

## A   Details of the GUI Agent

**Action Space.**   During the data processing stage, we standardized the action spaces separately for mobile and web domains to achieve unified representations suitable for joint training. In the evaluation stage, we further aligned actions across different benchmarks by mapping them to the corresponding standardized action spaces. The actions involved in both the training and evaluation phases fall within the categories detailed in Table 9 for the mobile domain and Table 10 for the web domain.

**Observation space.**   To simulate real-world human interactions with GUI environments and to better investigate whether middle-training data enhances the reasoning ability of GUI agents, the observation space in this study consists solely of visual information. Specifically, during task execution, GUI agents receive only visual feedback from the screen and historical action information, without leveraging any additional textual information from the environment (e.g., DOM or accessibility trees). Such textual information could introduce extra information gain, potentially confounding the effect of middle-training data.

## B   Details of Mid-Training Data

Vision-language data generally comprises paired visual and textual information, such as image captions, annotated screenshots, and visually grounded instructions. Considering the inherently multi-modal nature of GUI Agents, leveraging vision-language data may facilitate better alignment between visual and textual modalities, potentially improving agents' comprehension of and interaction with graphical interfaces. Our primary focus for vision-language modalities includes:

(1) **Chart/ Document Question Answering**: Data in this category enhance the model's ability to perform reasoning-intensive tasks over visual representations of struc-

tured charts and documents. Our training data is constructed by randomly sampling approximately 56K samples from InfographicVQA and Ureader QA in MAmmoTH-VL (Guo et al., 2024), 500 samples from MPDocVQA (Tito et al., 2023), and 93.5K samples from the warm-up data of MathV360k (Liu et al., 2024b). In total, this process yields a dataset of 150K samples for mid-training.

(2) **Non-GUI Perception**: This category enhances the model's perception capabilities for non-GUI images, such as posters, tables, and documents. Given the abundance of non-GUI perception datasets in online databases (e.g., documents (Kafle et al., 2018), posters (Ye et al., 2023), and plots (Methani et al., 2020)), we investigate whether leveraging such data can improve performance on GUI-related tasks. Specifically, we construct the training data by integrating 6.1K samples from Ureader OCR (Ye et al., 2023) with 143.9K randomly selected samples from DUE (Borchmann et al., 2021), yielding a total of 150k samples for the mid-training phase.

(3) **GUI Perception**: This category aims to enhance the model's perceptual capabilities for GUI images. To achieve this, the training dataset is constructed by randomly sampling 50K instances from each of the *Action Prediction*, *Webpage Question Answering*, and *Image Question Answering* subsets within MultiUI (Liu et al., 2024a), resulting in a total of 150K samples for the mid-training stage.

(4) **Multi-modal Math**: Math data (Cobbe et al., 2021; Shi et al., 2024) has been widely used to enhance the reasoning capabilities of LLMs and VLLMs. We explore whether incorporating multi-modal mathematical data can further enhance the planning capabilities of GUI-Agent. Specifically, we construct the training dataset by randomly sampling 150K multi-modal math problems from the Mavis dataset (Zhang et al., 2024b), which comprises high-quality mathematical questions accompanied by comprehensive reasoning processes.

(5) **Multi-round Visual Conversation**: Agent trajectories often consist of multiple steps, requiring the model to understand and memorize previous steps to inform current decisions. Multi-round visual conversation data exhibits similar characteristics, as generating a response in a given turn typically depends on the context of prior turns. We examine whether multi-turn multi-modal question-answering data can enhance the performance of GUI-Agent tasks. Specifically, we construct the training dataset by randomly sampling 150K multi-turn dialogues from the SVIT dataset (Zhao et al., 2023), which comprises multi-turn question-answering interactions involving intricate, image-based reasoning.

(6) **Web Screenshot2Code**: HTML code and corresponding web screenshots are readily available, providing rich information regarding the structure and interactivity of web elements (e.g., whether an icon is clickable or hoverable). We investigate whether leveraging such accessible data can enhance the performance of GUI-Agents on downstream tasks. Specifically, we construct the training dataset by randomly sampling 150K web screenshot-HTML code pairs from the Web2Code dataset (Yun et al., 2024).

(7) **Non-GUI Agent Trajectories**: With the rapid development of Embodied AI, a growing amount of non-GUI agent data is becoming available in domains beyond the web, such as household tasks (Shridhar et al., 2021). We investigate whether data from these domains can benefit GUI Agent tasks. Specifically, due to the limited availability of such data, we utilize all 51K samples from the *AlfWorld* subset of MAmmoTH-VL (Guo et al., 2024) as mid-training data for our experiments.

Text data is often more readily available and abundant compared to vision-language data, widely accessible through sources such as the internet. We investigate whether pure text data can enhance the capabilities of GUI Agents. For the text modality, our primary focus includes the following:

(1) **MathInstruct**: Text-based math datasets (Cobbe et al., 2021) are commonly used to enhance the reasoning capabilities of models. We randomly sample 150K examples from the *CoT* category of MathInstruct (Yue et al., 2023) as mid-training data.

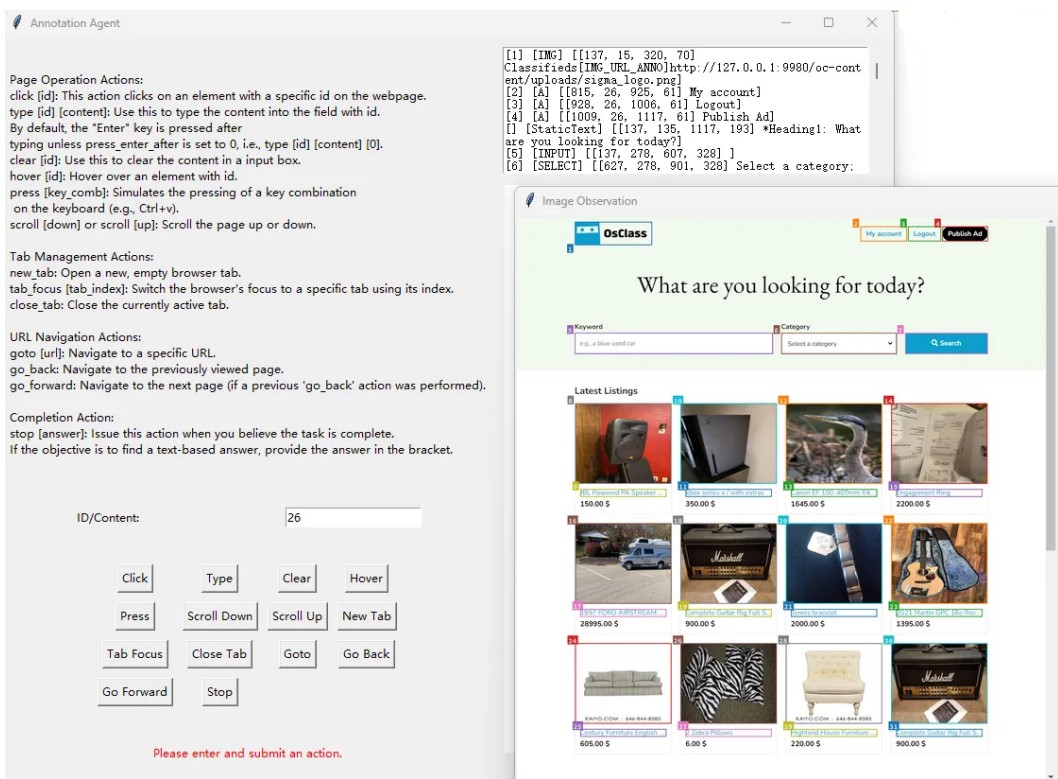

Figure 5: The annotation UI for VisualWebArena.

(2) **CodeI/O**: CodeI/O (Li et al., 2025) is a novel approach that transforms code-based reasoning patterns into natural language formats to enhance the reasoning capabilities of Large Language Models. We randomly sample 150K examples from the CodeI/O dataset.

(3) **Web Knowledge Base**: Text-based trajectories can be synthesized using methods like (Ou et al., 2024; Xu et al., 2024a), which leverage tutorial knowledge. However, generating multi-modal data at scale remains challenging. In this work, we explore whether text-based trajectory data can benefit vision-based GUI agents. Specifically, we utilize 100K trajectories from the Synatra dataset (Ou et al., 2024) and randomly sample 50K trajectories from the AgentTrek dataset (Xu et al., 2024a), both of which are web-domain datasets that may potentially enhance GUI-Agent performance on web-based tasks.

(4) **Olympiad Math**: NuminaMath (LI et al., 2024) is a comprehensive mathematical dataset comprising a wide range of problems, including exercises from Chinese high school mathematics curricula as well as competition problems from US and international mathematics olympiads. We specifically select the most challenging subset problems categorized under *Olympiads* within NuminaMath, and randomly sample 150K examples from this subset to highlight the complexity inherent in advanced mathematical reasoning.

## C  Details of GUI Trajectory Data

High-quality GUI trajectory data is critical for enabling GUI agents to advance digital automation. However, due to constraints in application scenarios and annotation costs, existing high-quality trajectory data remains limited. To better adapt mid-trained models to GUI-specific tasks, we collected the following trajectory data:

(1) **OS-Genesis**: We leverage trajectory data from both mobile and web platforms provided by OS-Genesis (Sun et al., 2024b), which are automatically synthesized in reverse via an interaction-driven approach. To further improve the inference quality of these trajectories, we enhance them with step-wise reasoning using *gpt-4o-mini* (Hurst et al., 2024). Specifically, *gpt-4o-mini* generates five sets of Chain of Thought data for each step and optimized them to ensure consistency with the corresponding action. If inconsistencies persist, the corresponding data will be discarded. Ultimately, approximately 4K high-quality single-step trajectory samples are collected for the web platform and 5K for the mobile platform.

(2) **MM-Mind2Web**: MM-Mind2Web (Zheng et al., 2024a) is the multi-modal extension of the Mind2Web dataset (Deng et al., 2023), designed for developing and evaluating generalist web agents capable of following natural language instructions to complete complex tasks on arbitrary websites. This dataset aligns each HTML document with its corresponding webpage screenshot from the Mind2Web raw dump, enabling joint modeling of structural and visual web information. We also employed *GPT-4o-mini* to process the data, resulting in the CoT-enhanced MM-Mind2Web dataset with 21K annotated steps.

(3) **VisualWebArena**: We randomly sample questions from different task template of the VisualWebArena (Koh et al., 2024a), and annotate 3,264 steps of data using our implemented annotation tools (Figure 5). To ensure the annotation is correct, each annotated trajectory will be replayed and is required to pass the task.

(4) **Aguvis**: Aguvis is a synthetic dataset of high-quality GUI agent trajectories that builds upon and enhances existing open-source datasets such as AitW (Rawles et al., 2023), GUI Odyssey (Lu et al., 2024a) and GUIAct (Chen et al., 2024). While these prior datasets typically include high-level goals, observations, and grounded actions, Aguvis enriches them by incorporating detailed intermediate reasoning and low-level action instructions. Leveraging a VLM, Aguvis generates step-by-step inner monologues comprising observation descriptions, agent thoughts, and fine-grained action instructions, enabling more advanced multi-modal reasoning and planning for GUI agents. We randomly sample 22K single-step reasoning data from Aguvis to support post-training.

## D  Training Details

Following state-of-the-art works (Xu et al., 2024b; Sun et al., 2024b; Gou et al., 2025), we use Qwen2-VL-7B-Instruct (Wang et al., 2024a) as our backbone model. During the mid-training experiments, the middle training stage and post-training stage are integrated under a single optimizer and learning rate schedule, which we find important empirically to stabilize the training process. For the scaling law experiments presented in Figure 3, we save checkpoints of the models trained on GUIMid as shown in Table 3 based on the amount of mid-training data trained. To evaluate the performance at intermediate points in the scaling law experiments (e.g., after training on 50K samples of mid-training data), we use the corresponding checkpoints and continue training with a cosine learning rate schedule that gradually reduces the learning rate to zero, starting from the learning rate value recorded at the checkpoint. Due to hardware constraints, we limit the batch size per device to 1 with a gradient accumulation of 4. For the results of scaling to 300K samples, we directly report the performance in Table 3.

## E  Evaluation Details

**AndroidWorld.** AndroidWorld evaluates autonomous agents in Android environments through 116 tasks across 20 real-world applications. Each task incorporates randomized parameters to generate diverse scenarios, enabling rigorous evaluation. Success rates (SR) are determined through system state inspection without modifying the original application source code. Due to application availability constraints, our final evaluation encompasses 111 tasks. We build GUI agents based on the M3A agent from AndroidWorld (Rawles et al.,

| Parameter | Value |
|---|---|
| Context Length | 8192 |
| Number of GPUs | 8 |
| Learning Rate | $2 \times 10^{-5}$ |
| Training Epochs | 1.0 |
| Batch Size Per Device | 2 |
| Gradient Accumulation | 2 |
| Learning Rate Scheduler | cosine |
| Warmup Ratio | 0.05 |
| Precision | bf16 |

Table 11: The training hyperparameters used in Table 3.

2024), drawing inspiration from the implementation of SeeAct-V agents (Gou et al., 2025), which rely solely on visual information by removing the SoM images and textual lists of elements from accessibility trees in the observations and converting element-based actions into pixel-level actions. Furthermore, to ensure the fairness and rigor of the experiment, we maintain consistency between the training and evaluation prompts by refraining from using deliberately crafted prompts targeting corner cases. This suggests that variations in agent performance can predominantly be attributed to the mid-training data.

**WebArena.** We utilize the agentboard (Ma et al., 2024) version of WebArena (Zhou et al., 2023), which contains 245 cases sampled from the original benchmark, enhanced with progress labels that provide both success rates and progress rates—offering more granular insights into model capability development. To ensure evaluation accuracy and consistency, we deploy the Docker-based websites on our local infrastructure rather than using the public AWS setup, which can lead to cross-evaluation inaccuracies. Map-related tasks were excluded due to website configuration issues, leaving 211 tasks. For maximum reliability, we restart the environment after each evaluation to eliminate potential cross-contamination between test sessions.

**gpt-4o Baselines** In the absence of 7B-parameter baselines with comparable data volume, we evaluate the strong model gpt-4o-2024-11-20 as a planning model in pure vision-based GUI scenarios, while employing UGround-V1-7B (Gou et al., 2025) as the grounding model. To ensure fair comparison, all models generate outputs using an identical prompt as shown in Figure 10,11. Notably, SeeAct-V (Gou et al., 2025) demonstrates exceptional performance under these same experimental conditions. This superior performance can be attributed to its meticulously crafted prompt specifically designed for AndroidWorld, which addresses common edge cases and incorporates rules aligned with human intuition.

All the experiments are conducted with temperate as 0.0, $top\_p$ as 1.0, max context length 8192.

# F  Prompt

We list our prompt below:

1. The prompt for generating Chain-of-Thought on OS-Genesis (Mobile) trajectories.
2. The prompt for generating Chain-of-Thought on OS-Genesis (Web) trajectories.
3. The prompt for generating Chain-of-Thought on VisualWebArena trajectories.
4. The prompt for generating Chain-of-Thought on Mind2Web trajectories.
5. The prompt for evaluation on the AndroidWorld.
6. The prompt for evaluation on the WebArena.

```
Prompt for Generating CoT on OS-Genesis (Mobile) Trajectories

You are a mobile agent.  Please think step by step and perform a series of actions
on an Android device to complete the task. At each stage, you will receive the current
screenshot, a record of previous actions, and a hint for the next correct step. Based on
this information, decide the next action to take without mentioning the provided correct
answer hint.

## Available actions:

### UI Operations:
- `click [element]`: Click on an element.
- `type [element] [value]`: Type content into a field by ID.
- `scroll [element] [value]`: Scroll the page or a specific element. The direction can be
`up`, `down`, `left`, or `right`. Leave the element blank for scrolling the entire page.
- `go_back`: Navigate back.
- `go_home`: Navigate to the home screen.
- `long_press [element]`: Long press on an element.
- `enter`: Press the Enter key.
- `open_app [app_name]`: Open an app by name.
- `wait [value]`: Wait for the screen to update for [value] seconds.

### Task finishing:
- `stop [value]`: Stop the task with a goal status or answer.  If you think the task
is completed or infeasible, set the status to `success` or `infeasible`.  If the task
requires an answer, provide the answer here.

### Instruction for the thought process:
1. Describe the situation in detail, focusing on the goal and the related visual cues
in current screenshot. Ensure your reasoning aligns with the goal, predicting the most
suitable action based on the screenshot and previous actions.
2. Aim to reason through the task as if solving it, rather than simply reflecting on the
labeled outcome.
3. Conclude the action with the format below.

Finally, end your thinking process with the action below.
In summary, the next action is:

```
{
"Element Description": "Describe the element you want to interact with, including its
identity, type (e.g., button, input field, dropdown, tab), and any visible text. Keep the
description concise and under 30 words. If there are similar elements, provide details to
distinguish them.",
"Action": "Select an action from the following options: {click, type, scroll, go_back,
go_home, long_press, enter, open_app, stop}. Choose one; do not leave it blank.",
"Value": "Provide additional input based on the chosen action:
- For 'type': specify the input text.
- For 'scroll': specify the direction ("up", "down", "left", "right").
- For 'open_app': provide the app name in the format: app_name="the name of the app".
- For 'stop': provide "completed", "infeasible", or the required answer.
- For 'wait': provide the waiting time in seconds, in the format: seconds="5s".
- For other actions: leave this field empty."
}
```

### Input:

**Previous Actions**: {previous_actions}
**Task**: {intent}
**Correct Action Hint**: {correct_answer}
```

Figure 6: Prompt for generating Chain-of-Thought on OS-Genesis (Mobile) trajectories.

```
Prompt for Generating CoT on OS-Genesis (Web) Trajectories

Imagine that you are imitating humans performing web navigation for a task, step by step. At each stage,
you can see the webpage as humans do through a screenshot, and you know the previous actions based on recorded
history, the current screenshot, and meta information about the current website. You need to decide on the next
action to take.
I will provide you with the hint answer. Please do not mention the hint answer in your thought process and just
reason through the task as if solving it yourself, but make sure your answer is the same with the hint answer.

## Available actions:

### Web Operations:
- `click [element]`: Click on an element.
- `type [element] [value]`: Type content into a field by ID.
- `clear [element]`: Clear the content of an element.
- `hover [element]`: Hover over an element by ID.
- `press [value]`: Press a key combination (e.g., Ctrl+v).
- `scroll [down]` or `scroll [up]'`: Scroll the page.

### Tab Management:
- `new_tab`: Open a new tab.
- `tab_focus [tab_index]`: Switch to a specific tab.
- `close_tab`: Close the current tab.

### URL Navigation:
- `goto [url]`: Navigate to a URL.
- `go_back`: Go to the previous page.
- `go_forward`: Go to the next page.

### Task finishing:
- `stop [answer]`: Issue this action when you believe the task is complete.

### Instruction for the thought process:

1. Describe the situation in detail, focusing on the goal and the related visual cues in current screenshot. Ensure
your reasoning aligns with the goal, predicting the most suitable action based on the screenshot and previous
actions.
2. Aim to reason through the task as if solving it, rather than simply reflecting on the labeled outcome.
3. Conclude the action with the format below.
4. Do not mention the hint answer in your thought process. Instead, reason to the answer independently, but ensure
your answer matches the hint answer.

Finally, end your thinking process with the action below.
In summary, the next action is:

```
{
"Element Description": "Describe the element you want to interact with, including its identity, type (e.g., button,
input field, dropdown, tab), and any visible text. Keep the description concise and under 30 words. If there are
similar elements, provide details to distinguish them.",
"Action": "Select an action from the following options: {stop, click, type, scroll, go_back, go_forward, goto,
clear, hover, press, new_tab, page_focus, close_tab}. Choose one action; do not leave this field blank.",
"Value": "Provide additional input based on the chosen action:
- For 'click': specify the element to click.
- For 'type': specify the input text.
- For 'scroll': specify the direction ("up", "down").
- For 'goto': specify the URL to navigate to.
- For 'clear': leave this field empty.
- For 'hover': specify the element to hover over.
- For 'press': specify the key combination to press.
- For 'tab_focus': specify the tab index to switch to.
- For 'stop': provide one of the following: "completed", "infeasible", or the required answer.
- For 'wait': provide the waiting time in seconds, in the format: seconds="5s".
- For all other actions: leave this field empty."
}
```

### Input:

**Previous Actions**: {previous_actions}

**Task**: {intent}

**Correct Action Hint**: {correct_answer}
```

Figure 7: Prompt for generating Chain-of-Thought on OS-Genesis (Web) trajectories.

```
Prompt for Generating CoT on VisualWebArena Trajectories

Imagine that you are imitating humans performing web navigation for a task, step by step. At each stage,
you can see the webpage as humans do through a screenshot, and you know the previous actions based on recorded
history, the current screenshot, and meta information about the current website. You need to decide on the next
action to take.
I will provide you with the hint answer. Please do not mention the hint answer in your thought process and just
reason through the task as if solving it yourself, but make sure your answer is the same with the hint answer.

## Available actions:

### Web Operations:
- `click [element]`: Click on an element.
- `type [element] [value]`: Type content into a field by ID.
- `clear [element]`: Clear the content of an element.
- `hover [element]`: Hover over an element by ID.
- `press [value]`: Press a key combination (e.g., Ctrl+v).
- `scroll [down]` or `scroll [up]`: Scroll the page.

### Tab Management:
- `new_tab`: Open a new tab.
- `page_focus [tab_index]`: Switch to a specific tab.
- `close_tab`: Close the current tab.

### URL Navigation:
- `goto [url]`: Navigate to a URL.
- `go_back`: Go to the previous page.
- `go_forward`: Go to the next page.

### Task finishing:
- `stop [answer]`: Issue this action when you believe the task is complete.

### Instruction for the thought process:
1. Describe the situation in detail, focusing on the goal and the related visual cues in the current screenshot.
Ensure your reasoning aligns with the goal, predicting the most suitable action based on the screenshot and previous
actions.
2. Aim to reason through the task as if solving it, rather than simply reflecting on the labeled outcome.
3. Conclude the action with the format below.
4. Do not mention the hint answer in your thought process, but make sure your answer is the same with the hint answer.

Finally, end your thinking process with the action below.
In summary, the next action is:

```
{
"Element Description": "Describe the element you want to interact with, including its identity, type (e.g., button,
input field, dropdown, tab), and any visible text. Keep the description concise and under 30 words. If there are
similar elements, provide details to distinguish them.",
"Action": "Select an action from the following options: {stop, click, type, scroll, go_back, go_forward, goto,
clear, hover, press, new_tab, page_focus, close_tab}. Choose one action; do not leave this field blank.",
"Value": "Provide additional input based on the chosen action:
- For 'click': specify the element to click.
- For 'type': specify the input text.
- For 'scroll': specify the direction ("up", "down").
- For 'goto': specify the URL to navigate to.
- For 'clear': leave this field empty.
- For 'hover': specify the element to hover over.
- For 'press': specify the key combination to press.
- For 'page_focus': specify the tab index to switch to.
- For 'stop': provide one of the following: "completed", "infeasible", or the required answer.
- For 'wait': provide the waiting time in seconds, in the format: seconds="5s".
- For all other actions: leave this field empty."
}
```

### Input:

Current URL: {url}

Previous Actions: {previous_actions}

Task: {intent}

Hint_Action: {hint_action}
```

Figure 8: Prompt for generating Chain-of-Thought on VisualWebArena trajectories.

```
Prompt for generating CoT on Mind2Web trajectories

You are a smart and helpful visual assistant that is well-trained to manipulate
websites. Your task is to navigate and take action on the current screen step-by-step to
complete the user request.
I will provide you with the hint answer. Please do not mention the hint answer in your
thought process and just reason through the task as if solving it yourself, but make sure
your answer is the same with the hint answer.

## Instructions:
- You will be provided with screenshots and website information.
- Review your previous actions to determine the next steps. Go back to previous status if
necessary.
- Pay close attention to all specific requirements of the task.

## Analysis Guidelines
### Previous Actions Analysis:
- You should analyze the previous actions and the current status of the task.
### Screenshot Description:
- You should describe all the screenshot in detail, especially the interactive elements,
such as buttons, search bars, and dropdown lists.
### Sub-task Planning:
- Analyze the task status based on the observation and past actions and detail a reasonable
future action plan to accomplish the user request.
- You should carefully check **ALL THE SPECIFIC REQUIREMENTS** to make the plan.
- You MUST check whether the last action is conducted successfully by analyzing the current
screenshot.
### Critical Analysis and Reflection:
- Check whether the history actions have accomplished the user request.
- Critique the past actions and make a decision on the next action, and decide whether to
backtrack to the previous steps with actions like: go back, goto ùrl; scroll ùp;
- Assess the feasibility of the current sub-task and the overall task, and decide whether
to modify the plan.

Finally, end your thinking process with the action below.
In summary, the next action is:

```
{
"Element Description": "Describe the element you want to interact with, including its
identity, type (e.g., button, input field, dropdown, tab), and any visible text. Keep the
description concise and under 30 words. If there are similar elements, provide details to
distinguish them.",
"Action": "Select an action from the following options: {click, type, scroll, go_back,
go_home, long_press, enter, open_app, stop}. Choose one; do not leave it blank.",
"Value": "Provide additional input based on the chosen action:
- For 'type': specify the input text.
- For 'scroll': specify the direction ("up", "down", "left", "right").
- For 'open_app': provide the app name in the format: app_name="the name of the app".
- For 'stop': provide "completed", "infeasible", or the required answer.
- For 'wait': provide the waiting time in seconds, in the format: seconds="5s".
- For other actions: leave this field empty."
}
```

### Input:

Task: {task}

Previous Actions: {previous actions}

Hint Answer: {hint_answer}
```

Figure 9: Prompt for generating Chain-of-Thought on Mind2Web trajectories.

```
Evaluation Prompt for Mobile Tasks

<image>
You are a mobile agent. You need to perform a series of actions to complete a task on
Android, step by step. At each step, you are provided with the current screenshot and
previous actions you have taken. You need to decide on the next action to take.

## Available actions:

### UI Operations:
- `click [element]`: Click on an element.
- `type [element] [value]`: Type content into a field by ID.
- `scroll [element] [value]`: Scroll the page or a specific element. The direction can be
'up', 'down', 'left', or 'right'. Leave the element blank for scrolling the entire page.
- `go_back`: Navigate back.
- `go_home`: Navigate to the home screen.
- `long_press [element]`: Long press on an element.
- `enter`: Press the Enter key.
- `open_app [app_name]`: Open an app by name.
- `wait [value]`: Wait for the screen to update for [value] seconds.

### Task finishing:
- `stop [value]`: Stop the task with a goal status or answer. If you think the task is
completed or infeasible, set the status to 'successful' or 'infeasible'. If the task
requires an answer, provide the answer here.

Please provide your detailed thought process and specify the action you intend to
perform. The action should include a description of the element to be operated on, the
type of action, and the corresponding value, formatted as follows:

```
{
"Element Description": "Describe the element you want to interact with.",
"Action": "Select an action from the available options, do not leave it blank.",
"Value": "Provide a value only if the action requires it."
}
```

### Input:

Previous Actions: {previous_actions}

Task: {intent}
```

Figure 10: Evaluation prompt for mobile tasks.

---

**Evaluation Prompt for Web Tasks**

\<image\>
Imagine that you are imitating humans performing web navigation for a task, step by step. At each stage, you can see the webpage as humans do through a screenshot, and you know the previous actions based on recorded history, the current screenshot, and meta information about the current website. You need to decide on the next action to take.

## Available actions:

### Web Operations:
- `click [element]`: Click on an element.
- `type [element] [value]`: Type content into a field by ID.
- `clear [element]`: Clear the content of an element.
- `hover [element]`: Hover over an element by ID.
- `press [value]`: Press a key combination (e.g., Ctrl+v).
- `scroll [down]` or `scroll [up]`: Scroll the page.

### Tab Management:
- `new_tab`: Open a new tab.
- `page_focus [tab_index]`: Switch to a specific tab.
- `close_tab`: Close the current tab.

### URL Navigation:
- `goto [url]`: Navigate to a URL.
- `go_back`: Go to the previous page.
- `go_forward`: Go to the next page.

### Task finishing:
- `stop [answer]`: Issue this action when you believe the task is complete (the value could be successful, infeasible, or the required answer).

Please provide your detailed thought process and specify the action you intend to perform. The action should include a description of the element to be operated on, the type of action, and the corresponding value, formatted as follows:

```
{
"Element Description": "Describe the element you want to interact with.",
"Action": "Select an action from the available options. Choose one; do not leave it blank.",
"Value": "Provide a value only if the action requires it."
}
```

### Input:

**Current URL**: {url}

**Previous Actions**: {previous_actions}

**Task**: {intent}

Figure 11: Evaluation prompt for web tasks.

