# OpenReview forum: "Breaking the Data Barrier -- Building GUI Agents Through Task Generalization"
_colmweb.org/COLM/2025/Conference — COLM 2025_

### Official Review · Reviewer_L7Ty · 2025-05-03

**Rating:** 6
**Confidence:** 3
**Ethics Flag:** 1

**Summary:**

Graphical User Interface (GUI) agents have recently become a hot topic, but high-quality GUI trajectory data remains scarce. This paper investigates the mid-training process of vision-language models (VLMs) to enhance GUI performance after specific fine-tuning. The study focuses on purely visual GUIs, which are easier to generalize. The authors collect 11 diverse non-GUI datasets — 7 multimodal and 4 textual — to conduct mid-training using Qwen2-VL-7B-Instruct. During mid-training, downstream GUI trajectory data is also incorporated. Experimental results on WebArena and AndroidWorld demonstrate that non-GUI mid-training significantly boosts final GUI performance, particularly with mathematical reasoning data. By identifying the most effective non-GUI samples, the authors construct the GUIMid dataset for combined mid-training.

**Reasons To Accept:**

The scarcity of high-quality GUI trajectory data is indeed a significant challenge. The use of 11 diverse non-GUI datasets for mid-training is interesting, and the experimental results show clear improvements on GUI benchmarks.

**Reasons To Reject:**

The mid-training process appears to serve as a general form of data augmentation rather than being specifically tailored to GUI tasks. Moreover, the experiments are conducted solely with Qwen2-VL-7B-Instruct, raising concerns about the generalizability of the results, as different VLMs may respond differently to mid-training due to variations in pre-training data and data ratios.

---

> ### Author Response · Authors · 2025-06-03
> **Rebuttal by Authors**
>
> Thanks very much for your valuable suggestions and positive feedback! We address your concerns below.
>
> > **Q1: The mid-training process appears to serve as a general form of data augmentation rather than being specifically tailored to GUI tasks.**
>
> A1: Thank you for the feedback. Indeed, the motivation of this paper is to explore the generalization of general-form, non-GUI mid-training on GUI tasks.
>
>
> > **Q2: the experiments are conducted solely with Qwen2-VL-7B-Instruct.**
>
> A2: Thanks for your valuable suggestions. We conducted additional experiments using a stronger multi-modal backbone to validate our findings. Specifically, we employed `gemma-3-12b-it`[1] as the base model. The experimental setup remains identical to Table 3, with only the backbone model changed. We report the Progress Rate (PR) and Success Rate (SR) below:
>
>
> | **Training Strategy** | **WebArena (PR)** | **WebArena (SR)** | **AndroidWorld (SR)** |
> |:----------------------:|:-----------------:|:-----------------:|:---------------------:|
> | GUI Post-Training Only | 29.9 | 9.5 | 13.5 |
> | Mid-Training + GUI Post-Training | **32.6** | **10.0** | **18.5** |
>
> The results demonstrate consistent improvements on `gemma-3-12b-it`, with our mid-training approach achieving +2.7 PR, +0.5 SR on WebArena and +5.0 SR on AndroidWorld. This indicates the generalizability of our findings beyond the original Qwen2-VL backbone.
>
> We will include these experiments in the revised version of our paper.
>
> [1] Team, Gemma, et al. "Gemma 3 technical report." arXiv preprint arXiv:2503.19786 (2025).

---

> > ### Comment · Reviewer_L7Ty · 2025-06-09
> > **Acknowledgment of Rebuttal**
> >
> > Thank you to the authors for their response, which partially addresses my concerns. However, I remain concerned about the generalization and therefore maintain my score.

---

> > > ### Author Response · Authors · 2025-06-11
> > > **Thank you for your response**
> > >
> > > Thanks for your valuable feedback. We use public datasets, which may have potentially been trained by VLMs. But we re-enhanced the capabilities brought by this data during the mid-training stage. The training order has a significant impact.
> > >
> > > And we provide insights on what kind of non-GUI data are helpful during the mid-training stage, which can give the community new inspiration for data collection directions, not just specific datasets.

---

### Official Review · Reviewer_yoKR · 2025-05-10

**Rating:** 7
**Confidence:** 4
**Ethics Flag:** 1

**Summary:**

The paper tackles the challenge of limited high-quality trajectory data for training GUI agents. Specifically, the authors propose to introduce a mid-training stage using VLMs trained on a diverse set of reasoning-intensive tasks—such as GUI perception, multimodal reasoning, and text-based reasoning—to improve generalization to GUI planning. Experiments across 11 tasks reveal that task generalization is highly effective, with notable cross-modal transfer benefits (e.g., text-only math data improving visual GUI tasks), while GUI perception data alone yields limited benefits. By combining the most effective mid-training tasks with optimized mixture, the authors achieve significant performance gains of 8.0% on WebArena and 12.2% on AndroidWorld.

**Questions To Authors:**

- How were the 11 mid-training tasks selected? Were there specific criteria for inclusion or exclusion?
- Why do the authors think text-only math reasoning tasks generalize so effectively to visual GUI planning? Is there a shared reasoning structure that enables this?

**Reasons To Accept:**

- Novel empirical contribution: This work is the first to systematically show that reasoning-intensive data from other domains can significantly enhance GUI planning, offering a practical solution for a field where collecting high-quality training data is particularly expensive.
- Insightful findings: The work provides valuable observations for the community, notably that text-only math data yields the largest gains for visual GUI tasks, while GUI perception data—despite its apparent relevance—offers limited benefits.
- Strong empirical results: Extensive experiments validate the proposed mid-training strategy, showing that optimized mixture tasks lead to substantial improvements on challenging benchmarks, with gains of 8.0% on WebArena and 12.2% on AndroidWorld.

**Reasons To Reject:**

* Limited novelty in methodology: While the findings are insightful, the core approach—leveraging mid-training on a mixture of auxiliary tasks—is conceptually incremental and builds on existing transfer learning paradigms without introducing fundamentally new techniques.  Moreover, mid-training has become a standard stage in modern LLM training.

---

> ### Author Response · Authors · 2025-06-03
> **Rebuttal by Authors**
>
> Thanks very much for your valuable suggestions and positive feedback! We address the concerns below:
>
> > **Q1: Limited novelty in methodology**
>
> A1: We would like to clarify that **our contributions are fundamentally data-centric rather than algorithmic**, introducing two key innovations: (1) we are the first to propose to leverage data-rich non-GUI domains to enhance foundational abilities for data-scarce GUI tasks, and (2) Empirical insights demonstrating how non-GUI domain data can effectively improve GUI agent performance. These findings provide the community with actionable guidance for rethinking GUI agent training data construction strategies.
>
> > **Q2: How were the 11 mid-training tasks selected? Were there specific criteria for inclusion or exclusion?**
>
> A2: We select the 11 mid-training tasks based on the core capabilities required for effective GUI agents: (1) Perception and Understanding and (2) Reasoning and Planning [1]. As shown in Table 1, we targeted four specific abilities that support these core capabilities: perception, reasoning, interaction, and knowledge.
>
> Our selection criteria were:
> - Hypothesized capability alignment: Each task was hypothesized to enhance one of the four fundamental abilities, with effectiveness verified experimentally
> - Data availability: We prioritize domains with abundant, high-quality datasets
> - Modality exploration: Given that text-only data is more abundant and often higher quality than multimodal data, we investigate whether pure text tasks could benefit GUI agents
>
>
> [1] Xu, Yiheng, et al. "Aguvis: Unified Pure Vision Agents for Autonomous GUI Interaction." ICML 2025
>
>
> > **Q3: Why do the authors think text-only math reasoning tasks generalize so effectively to visual GUI planning?**
>
> A3: We hypothesize that text-only math reasoning tasks enhance GUI planning through two key mechanisms:
> (1) Sequential reasoning alignment: GUI agents must analyze past action trajectories (typically text-based) and reason about subsequent actions. Math reasoning tasks strengthen this sequential logical reasoning capability, which directly transfers to GUI planning scenarios.
> (2) Language component enhancement: Multimodal foundation models may experience degradation in their language reasoning components during multimodal training [1]. Text-only math tasks specifically target and strengthen the language reasoning abilities within these models, compensating for potential capability loss
>
>
> [1] Wang, Weihan, et al. "Cogvlm: Visual expert for pretrained language models." Advances in Neural Information Processing Systems 37 (2024): 121475-121499

---

> > ### Comment · Reviewer_yoKR · 2025-06-08
> > **Acknowledgment of Rebuttal**
> >
> > After reviewing the authors' responses, I maintain my positive assessment of the work.

---

> > > ### Author Response · Authors · 2025-06-11
> > > **Thank you for your response**
> > >
> > > Thank you for your response and again for your valuable suggestions !

---

### Official Review · Reviewer_GCJU · 2025-05-13

**Rating:** 3
**Confidence:** 5
**Ethics Flag:** 1

**Summary:**

The paper investigate the challenge of data scarcity for training agents that automate tasks on GUI. It proposes training these VLMs on data-rich, reasoning-focused tasks as a mid-training step to improve their ability to generalize to GUI planning scenarios. The research found that multimodal mathematical reasoning and text-only mathematical data significantly boosted performance on benchmarks like AndroidWorld and WebArena, demonstrating effective cross-modal generalization, while GUI perception data had limited impact.

**Reasons To Accept:**

The paper presents comprehensive experimental reports on GUI, demonstrating the effectiveness of the proposed mid-training approach and the GUIMid dataset through performance improvements.

**Reasons To Reject:**

1. Lack of technical depth: The paper largely functions as an experimental report on combining existing open-source datasets rather than introducing significant technical novelty.

2. Limited generalization: The conclusions regarding the optimal data mixture are strictly confined to performance on WebArena and AndroidWorld; the lack of evaluation on diverse GUI environments (e.g., OS World, Windows Agent Arena, WebVoyager) leaves the generalizability of the findings, particularly the proposed data composition, uncertain.

3. Misleading title: The title is somewhat misleading, implying a solution to the fundamental challenge of collecting scarce GUI data, when the core contribution appears to be an empirical report and evaluation of already available open-source datasets.

---

> ### Author Response · Authors · 2025-06-03
> **Rebuttal by Authors**
>
> > **Q1. Lack of technical depth**
>
> A1:  We would like to clarify that our novelty lies in **data-centric contributions rather than algorithmic innovations**:
>
> **(1) Cross-domain mid-training framework for GUI agents:** We are the first to systematically investigate and demonstrate how data-rich non-GUI domains can enhance the fundamental capabilities of GUI agents through mid-training. Our work pioneers cross-domain mid-training to bridge heterogeneous domains (mathematical, code, text knowledge) with GUI tasks, representing a novel methodological contribution in terms of data  to the GUI agent field. This is the first work to demonstrate that GUI agents can effectively leverage abundant non-GUI data to alleviate the data scarcity challenges inherent in GUI domains.
>
> **(2) Novel empirical insights on cross-domain transferability:** We provide crucial findings about which specific data types from non-GUI domains are transferable and beneficial for GUI tasks, as acknowledged by Reviewer yoKR as well. For example, our systematic analysis reveals that mathematical and code  reasoning significantly improve GUI planning capabilities, even the text-only data. These insights offer actionable, evidence-based guidance for the community's training data construction.
>
> > **Q2: Limited generalization**
>
>
>
> A2: Thank you for this valuable suggestion. We provide additional experiments: on interactive and  real-world benchmarks: Mind2Web-Live [1] and WebVoyager [2]. The results are shown in the Q2 of the General Response.
>
> From Tables 1 and 2, our method demonstrates significant improvements in interactive real-world web environments. On Mind2Web-Live, our approach achieves a 67% relative improvement (10.2 vs 6.1). On WebVoyager, we observe an 83% relative improvement (37.9 vs 20.7) on average, with particularly strong gains in domains requiring complex reasoning such as Cambridge Dictionary (+250%) and Wolfram Alpha (+150%). These results indicate that our mid-training approach enhances fundamental capabilities that transfer effectively to diverse, dynamic web environments.
>
> We will include these experiments in the revised version of our paper.
>
>
> [1] Pan, Yichen, et al. "WebCanvas: Benchmarking Web Agents in Online Environments." Agentic Markets Workshop at ICML 2024.
> [2] He, Hongliang, et al. "WebVoyager: Building an End-to-End Web Agent with Large Multimodal Models." Proceedings of the 62nd Annual Meeting of the Association for Computational Linguistics (Volume 1: Long Papers). 2024.
>
>
> > **Q3: Misleading title**
>
> A3:  We respectfully disagree that our title is misleading. Our work aims to build stronger GUI agent models by mid-training on non-GUI agent domains to benefit GUI tasks. The contribution lies in a **new mid-training task generalization strategy** for GUI agents, which is accurately reflected in our title: “Building GUI Agents Through Task Generalization”.

---

> > ### Comment · Reviewer_GCJU · 2025-06-09
> >
> > Regarding "Cross-domain mid-training framework": This is indeed a well-established method in LLM training, not a novel contribution. Simply applying existing mid-training method to GUI agents constitutes application rather than innovation. The claim of being "the first to systematically investigate" cross-domain mid-training for GUI agents doesn't constitute a technical contribution, it's merely domain-specific application of existing methods.
> >
> > Web-centric Evaluation Bias: Your additional experiments on Mind2Web-Live and WebVoyager still confine evaluation to web environments. This narrow scope severely limits the generalizability claims of your work.
> >
> > Missing Desktop GUI Evaluation: The absence of evaluation on desktop-level GUI tasks like OSWorld is a critical oversight. Desktop GUI interactions involve fundamentally different interaction paradigms, application contexts, and complexity levels compared to web interfaces. Without demonstrating effectiveness across diverse GUI modalities, your claims of cross-domain transferability remain unsubstantiated.
> >
> > Cherry-picked Domains: The highlighted improvements in "Cambridge Dictionary" and "Wolfram Alpha" appear to be cherry-picked examples that align with your training data domains (text and mathematical reasoning), which doesn't demonstrate true cross-domain generalization.
> >
> > If your approach truly represents a breakthrough in GUI agent capabilities, why limit evaluation to web environments where the interaction patterns are relatively constrained? Desktop GUI tasks would provide a more rigorous test of whether your cross-domain mid-training actually enhances fundamental reasoning capabilities or merely improves performance on similar web-based tasks.
> >
> > The work appears to be an incremental application of existing methods rather than a fundamental contribution to the field.

---

> > > ### Author Response · Authors · 2025-06-11
> > > **Authors’ Response – Round 2 (1/2)**
> > >
> > > Thanks for your response. We continue to address your concerns about web and mobile domains not being representative of GUI and the novelty of our work.
> > >
> > > > **Q1: Without demonstrating effectiveness across diverse GUI modalities, your claims of cross-domain transferability remain unsubstantiated.**
> > >
> > > GUIs span multiple platforms including XR, web, mobile, and desktop. Web and mobile platforms constitute core domains with substantial representation in GUI interaction research. Our evaluation setting is consistent with many previous works [1, 2, 3, 4, 5, 6] that validate their GUI agents specifically on Web and Android domains, while others focus exclusively on desktop environments [7]. While evaluating on all GUI categories is more comprehensive, we argue that our evaluation settings already cover multiple benchmarks and are consistent with most previous works. We do understand the reviewer’s concerns on generalizing to desktop GUIs, we will clarify the scope of our claims as “cross-domain transferability to web and mobile GUI environments” in the next revision of the paper.
> > >
> > >
> > >
> > >
> > >
> > > > **Q2: Missing Desktop GUI Evaluation:**
> > >
> > >
> > >
> > > First, as discussed in Q1 above, we think our evaluations on multiple web and mobile benchmarks are representative and the settings are consistent with many previous works, most of which did not evaluate on all GUI categories. Second, we would like to explain why we did not report the results on desktop:
> > >
> > > (1) **Action space incompatibility**: Due to the inconsistent action spaces between Mobile/Web and desktop platforms, we could not directly test our trained models on desktop during the rebuttal period, which requires retraining.
> > >
> > > (2) **Desktop evaluation challenges**: Due to desktop-specific difficulties (particularly the lack of high-quality desktop datasets before our paper submission), currently except for models based on UI-TARS [8], no open-source 7B-scale models serve as planners in the OS-World leaderboard [9]. Researchers typically evaluate 7B models on desktop by using commercial models (e.g., GPT-4o) as planners and 7B-scale models for grounding. When using self-trained models as the planner, they normally use at-least 72B-scale models [10][11]. Constrained by computational limitations, we could not train 72B-scale models due to the different action spaces involved. Therefore, it is common that 7B planners are not benchmarked in OS-World environments due to the challenges.
> > >
> > >
> > > [1] Hong, Wenyi, et al. "CogAgent: A Visual Language Model for GUI Agents." 2024 IEEE/CVF Conference on Computer Vision and Pattern Recognition (CVPR). IEEE, 2024.
> > > [2] Liu, Xiao, et al. "Autoglm: Autonomous foundation agents for guis." arXiv preprint arXiv:2411.00820 (2024).
> > > [3] Sun et al. OS-Genesis: Automating GUI Agent Trajectory Construction via Reverse Task Synthesis. ACL 2025 main.
> > > [4] Bai, Hao, et al. "Digirl: Training in-the-wild device-control agents with autonomous reinforcement learning." Advances in Neural Information Processing Systems 37 (2024): 12461-12495.
> > > [5] Liu, Yuhang, et al. "InfiGUIAgent: A Multimodal Generalist GUI Agent with Native Reasoning and Reflection." arXiv preprint arXiv:2501.04575 (2025).
> > > [6] Ziyang, Meng, et al. "VGA: Vision GUI Assistant-Minimizing Hallucinations through Image-Centric Fine-Tuning." Findings of the Association for Computational Linguistics: EMNLP 2024. 2024.
> > > [7] Lin, Kevin Qinghong, et al. "VideoGUI: A Benchmark for GUI Automation from Instructional Videos." The Thirty-eight Conference on Neural Information Processing Systems Datasets and Benchmarks Track.
> > > [8] Qin, Yujia, et al. "UI-TARS: Pioneering Automated GUI Interaction with Native Agents." arXiv preprint arXiv:2501.12326 (2025).
> > > [9] Xie, Tianbao， et al. https://os-world.github.io/
> > > [10] Xu, Yiheng, et al. "Aguvis: Unified Pure Vision Agents for Autonomous GUI Interaction." ICML 2025
> > > [11] Xie, Tianbao, et al. "Scaling Computer-Use Grounding via User Interface Decomposition and Synthesis." arXiv preprint arXiv:2505.13227 (2025).

---

> ### Author Response · Authors · 2025-06-11
> **Authors’ Response – Round 2 (2/2)**
>
> > **Q3: it's merely domain-specific application of existing methods**
>
> Our contribution is fundamentally **data-centric**, while the major challenge in GUI agent training is in limited data: 1. Data collection is expensive. 2. Collected agent trajectories are not diverse enough, and leads to saturated scaling. This paper reveals which non-GUI data are helpful to transfer to our evaluated GUI tasks – these insights are new, and **not existing**.
> Speaking of “methods” or “technical contribution”, if the reviewer is referring to an innovative training or data collection approach, it is true that we don’t have it – we never claim it,  and we don’t see any problem with that. Empirical contribution and gaining insights from empirical experiments align with the objectives of COLM and are pretty important in LLM research.
>
> > **Q4: Cherry-picked Domains**
>
> Our Q2 in the General Response demonstrated significant performance gains on 12 out of 14 domains.  We evaluated and reported results on **all WebVoyager domains except Google Search (due to reCAPTCHA limitations)** instead of picking just two domains. In the text description, we did highlight two domains where we performed well, but **we reported performance on all domains rather than hiding anything, we disagree that this can be criticized as “cherry-picking”.**
> Moreover, it is important to note that our interactions with Wolfram Alpha and Cambridge Dictionary involve multimodal interations. Such  scenarios are fundamentally distinct from purely textual or mathematical domains. Thus, the significant performance improvement demonstrated is genuinely indicative of robust cross-domain generalization, rather than alignment or overfitting to familiar domains.

---

### Official Review · Reviewer_qdBH · 2025-05-14

**Rating:** 7
**Confidence:** 4
**Ethics Flag:** 1

**Summary:**

This paper proposes enhancing GUI agent generalization by mid-training Vision Language Models (VLMs) on diverse, reasoning-intensive tasks with abundant instruction-tuning data. It finds that multimodal and even text-only reasoning tasks significantly outperform GUI perception data, leading to strong cross-modal gains. The authors curate optimized training mixtures, achieving up to 8-12%  improvement on WebArena and AndroidWorld.

**Questions To Authors:**

It is interesting to see with just this kind of mid-training data, Qwn2-VL-7B model is performing significantly better. But since the main contribution of this paper is the study that such mid-training data helps in GUI tasks, I think it is important to have some more comprehensive experiments done

Other than Aguvis, I suggest the authors also consider comparison with other GUI models like UI_TARS, OS-ATLAS, InfiGUIAgent and older work like CogAgent, ShowUI etc. Or if there is any compelling reason not to compare with these, authors can provide their argument.
Compare on other benchmarks, Multimodal Mind2Web and Mind2Web Live,OSWorld, AndroidControl, GUIOdyssey, AITW, AITZ
Other than that, some more questions:

- From the scaling law figs of GUIMid Mid-Training data it seems that the performance on both benchmarks can further improve beyond 300K. Any reason why it was stopped at 300K?
- For the paper, it would be nice to do a case-study or analysis of what kinds of tasks are least or most benefited by these kinds of mid-training reasoning data.
- How much GUI trajectory data is mixed in the mid-training phase ? Is this added to each of the individual settings (i.e. each row) in the Table 3? Also, if that is the case, why isn’t the ‘MathInstruct (mixing)’ number of Table 5 not matching with the corresponding result with MathInstruct data in Table 3?
- By using the mid-training data, can the authors show that the model becomes more sample-efficient on the post-training data i.e. with lesser training data it achieves similar final performance. Also, if that’s the case would mid-trained models require lesser number of expensive post-training samples with planning or inner-monologue rich data like the 22k from Aguvis, to achieve similar performance?

**Reasons To Accept:**

- good improvement in results (8%-12% on WebArena and AndroidWorld)
- its an interesting study to observe that more textual and multimodal reasoning training data can help improve downstream GUI tasks.

**Reasons To Reject:**

- not much novelty other than using some standard datasets for training GUI models. But still it is an interesting and previously unknown (afaik) observation that these kinds of training can have a big impact on the GUI performance. However this is the main observation, doing it on more benchmarks is imp.

---

> ### Author Response · Authors · 2025-06-03
> **Rebuttal by Authors (1/2)**
>
> We thank the reviewer for the positive comments! We address the concerns below:
>
> > **Q1. not much novelty other than using standard datasets for training GUI models.**
>
> A1: We appreciate your feedback and would like to clarify that our novelty lies in **data-centric contributions rather than algorithmic innovations**:
>
> **(1) Cross-domain mid-training framework:** We are the first to propose using data-rich non-GUI domains to enhance the fundamental abilities of GUI agents and generalize to GUI tasks.
>
> **(2) Novel empirical insights:** As Reviewer yoKR noted, we provide crucial findings about which domain-specific data types are transferable and beneficial for GUI tasks, offering actionable guidance for the community's training data construction.
>
> These contributions advance understanding of effective GUI agent training beyond conventional domain-specific dataset usage.
>
>
>
> > **Q2. doing it on more benchmarks is imp**
>
> A2: Thank you for your valuable suggestion. We provide additional experiments on two interactive and  real-world benchmarks:  Mind2Web-Live [1] and WebVoyager [2]. The results are shown in the Q2 of the General Response.
>
> From Tables 1 and 2, our method shows significant improvements in interactive web environments: 67% relative improvement on Mind2Web-Live (10.2 vs 6.1) and 83% on WebVoyager (37.9 vs 20.7), with particularly strong gains in complex reasoning domains like Cambridge Dictionary (+250%) and Wolfram Alpha (+150%). These results indicate that our mid-training approach enhances fundamental capabilities that transfer effectively to diverse web environments.
>
> [1] Pan, Yichen, et al. "WebCanvas: Benchmarking Web Agents in Online Environments." Agentic Markets Workshop at ICML 2024.
> [2] He, Hongliang, et al. "WebVoyager: Building an End-to-End Web Agent with Large Multimodal Models." Proceedings of the 62nd Annual Meeting of the Association for Computational Linguistics (Volume 1: Long Papers). 2024.
>
>
> > **Q3. I suggest the authors also consider comparison with other GUI models like UI_TARS, OS-ATLAS, InfiGUIAgent and older work like CogAgent, ShowUI etc**
>
> A3: Thanks for your valuable suggestions. We report the Success Rate (SR) of UI_TARS and OS-Atlas on AndroidWorld. Due to time and resource constraints, we will report the results on WebArena during the discussion period. For OS-Atlas [1], we use `OS-Atlas-Pro-7B`(the end-to-end model). For UI-TARS [2], we evaluate its latest open-source end-to-end model, `UI-TARS-1.5-7B`.
>
> | **Domains** | **Observation** | **WebArena PR** | **WebArena SR** | **AndroidWorld SR** |
> |:-------------|:----------------:|:-----------------:|:-----------------:|:---------------------:|
> | **GUI Post-Training Only** | Image | 26.3 | 6.2 | 9.0 |
> | **Public Baselines** |  |  |  |  |
> | GPT-4o-2024-11-20 | Image | 36.9 | 15.6 | 11.7 |
> | OS-Genesis-7B | Image + Accessibility Tree | – | – | 17.4 |
> | AGUVIS-72B | Image | - | - | 26.1 |
> | Claude3-Haiku | Accessibility Tree | 26.8 | 12.7 | - |
> | Llama3-70b | Accessibility Tree | 35.6 | 12.6 | - |
> | Gemini1.5-Flash | Accessibility Tree | 32.4 | 11.1 | - |
> | OS-Atlas-Pro-7B | Image | To be done  | To be done | 0 |
> | UI-TARS-1.5-7B | Image | To be done  | To be done | 30.2 |
>
> While UI-TARS achieves the best performance, our primary contribution is the analysis of generalizable domains rather than pursuing the highest performance, and performance is also related to base models and training data scale.
>
> [1] Wu, Zhiyong, et al. "Os-atlas: A foundation action model for generalist gui agents." arXiv preprint arXiv:2410.23218 (2024).
> [2] Qin, Yujia, et al. "UI-TARS: Pioneering Automated GUI Interaction with Native Agents." arXiv preprint arXiv:2501.12326 (2025).
>
> > **Q4. it seems that the performance on both benchmarks can further improve beyond 300K. Any reason why it was stopped at 300K?**
>
> A4: Due to computational constraints, our main experiments use 300K samples, but we conduct additional experiments with 600K data to validate the scaling trend:
>
> | Training Strategy   | WebArena (Progress Rate) | WebArena (Success Rate) | AndroidWorld (Success Rate)|
> |:-------------------------------:|:--------------------------:|:-------------------------:|:--------------:|
> | GUI Post-Training Only        | 26.3                     | 6.2                     | 9.0          |
> | GUIMid (300k) + Post-Training | 34.3                     | 9.5                     | **21.2**   |
> | GUIMid (600k) + Post-Training | **35.7**                     | **11.9**                    |    **21.2**  |
>
> The results confirm consistent improvements with scaling: success rate increases from 9.5% to 11.9% (+25% relative improvement) in WebArena. However, AndroidWorld shows no change, possibly because performance is bounded by specific knowledge or reasoning capabilities that require different data compositions rather than simply more data.

---

> > ### Author Response · Authors · 2025-06-03
> > **Rebuttal by Authors (2/2)**
> >
> > > **Q5. it would be nice to do a case-study or analysis of what kinds of tasks are least or most benefited**
> >
> > Thank you for this suggestion. We conducted human analysis of reasoning trajectories from Table 3 models with/without mid-training. While we did not find clear task-level distinctions (likely because GUI tasks are multi-step and test various capabilities across steps), we identified key improvements in reasoning patterns after mid-training:
> >
> > **(1) Enhanced Action Result Analysis:**
> > Task: Record an audio clip using Audio Recorder app and save it.
> >
> > **With GUIMid + Post-Training:**
> > thought: ...The screenshot shows the setup options for recording, including the recording format, sample rate, bitrate, and channel count. *The next logical step is to finalize the recording settings by clicking the "Apply" button, which is also prominently displayed in yellow. This action will confirm the recording parameters and allow me to proceed with the audio recording.*
> >
> > **With Post-Training only:**
> > thought: ...The "Apply" button is also present, which is likely the next step to confirm the settings before recording. *Given that the next logical step is to confirm the settings and proceed with the recording, I will click the "Apply" button.*
> >
> > **(2) More Thorough Exploration Strategy:**
> >
> > Task: Delete the following recipes from Broccoli app: Thai Peanut Noodle Salad.
> >
> > **With GUIMid,** the model considers multiple possibilities, for example:
> >
> > In this situation, I have opened the Broccoli - Recipe App and ... *Typically, in recipe apps, there may be a button or icon that allows users to delete or manage recipes, often represented by a trash can, a heart, or a menu option. Since the screenshot shows the recipe details, I will look for any visible options that indicate a deletion function.
> > Given that I need to find a way to delete the recipe, I will look for a menu or options button
> > that might allow me to delete it*.
> >
> >
> > **Without GUIMid,** the model tends to default to simple scrolling actions without exploring alternative interaction patterns, for example:
> >
> > In the current situation, I have opened the Broccoli… Since the screenshot does not show any visible buttons or icons,*I will need to scroll the page to see if the delete option is located further down.*
> >
> > This aligns with Figure 2's case study, where mid-training enables more comprehensive reasoning about possible actions when target elements are not immediately visible. While the post-training only ones just keep scrolling down.
> >
> >
> > > **Q6.  How much GUI trajectory data is mixed in the mid-training phase?  Why isn’t the ‘MathInstruct (mixing)’ number of Table 5 not matching with the corresponding result with MathInstruct data in Table 3**
> >
> > A6: Thank you for pointing this out. During mid-training, we mixed all 55k post-training data into the mid-training phase.
> > The discrepancy between Table 5 and Table 3 (33.6 vs 31.9 on WebArena) occurs because GUI tasks involve real-time environment interactions, even with greedy decoding. The performance shows a variation of around 2% on WebArena and 3% on AndroidWorld between the maximum and minimum values, which is lower than our improvement. Furthermore, both results consistently demonstrate significant improvement over the baseline (26.3) without mid-training, confirming our approach's effectiveness.
> >
> > > **Q7. Can the authors show that the model becomes more sample-efficient on the post-training data? Would mid-trained models require lesser number of expensive post-training samples with planning or inner-monologue rich data**
> >
> > A7: Thank you for this question. Table 3 shows that with the same post-training data, the mid-trained model achieves higher success rates (9.5/21.2 vs 6.2/9.0 on web/mobile tasks), so we hypothesize the answer is yes. However, we have not conducted controlled experiments with varying post-training data sizes during the rebuttal period. We will include this analysis in the next.
> > [1] Xu, Yiheng, et al. "Aguvis: Unified Pure Vision Agents for Autonomous GUI Interaction." ICML 2025
> >
> > We will include these experiments in the revised version of our paper.

---

> > > ### Comment · Reviewer_qdBH · 2025-06-09
> > >
> > > Thank you to the authors for your response. In light of that i would like to maintain my positive rating

---

> > > > ### Author Response · Authors · 2025-06-11
> > > > **Thank you for the response**
> > > >
> > > > Thank you for your valuable feedback to help us refine the quality of our paper. We will further polish the paper in the final revision. Thank you!

---

> ### Author Response · Authors · 2025-06-10
> **Rebuttal by Authors**
>
> Thanks for your positive feedbacks!
>
> > **More experiments about Q7.  Would mid-trained models require lesser number of expensive post-training samples with planning or inner-monologue rich data**
>
>
>
> To investigate whether mid-trained models are more data-efficient during post-training, we controlled the amount of in-domain trajectory data used in both mid-training and post-training stages by randomly sampling different ratios of the original post-training dataset. The results show that our GUI Mid + Post-training approach achieves comparable performance using only 20% of post-training data (WebArena Progress Rate: 25.5 vs 26.3 baseline, Success Rate: 6.6 vs 6.2, AndroidWorld: 10.8 vs 9.0), and significantly outperforms the baseline with 60% data across all metrics (WebArena Progress Rate: 27.0 vs 26.3, Success Rate: 8.1 vs 6.2, AndroidWorld: 12.6 vs 9.0). These findings demonstrate that mid-training substantially reduces expensive post-training data requirements while maintaining or improving performance.
>
> | Training Strategy | Post-training Data Ratio | WebArena (Progress Rate) | WebArena (Success Rate) | AndroidWorld (Success Rate) |
> |-------------------|-------------------------|--------------------------|-------------------------|----------------------------|
> | GUI Post-Training Only | 100% | 26.3 | 6.2 | 9.0 |
> | GUI Mid + Post-training | 20% | 25.5 | 6.6 | 10.8 |
> | GUI Mid + Post-training | 60% | 27.0 | 8.1 | 12.6 |

---

> > ### Comment · Reviewer_qdBH · 2025-06-10
> >
> > Thank the authors for the additional experiments. I have increased my score to 7

---

### Author Response · Authors · 2025-06-03
**General Response - Contribution Declaration and Additional Benchmarks (1/2)**

We thank the reviewers for their constructive feedback. We clarify our key contributions and justify our evaluation methodology on AndroidWorld and WebArena, while providing additional results on WebVoyager and Mind2Web-Live benchmarks to address concerns regarding novelty and evaluation comprehensiveness.

**Q1:Novelty on empirical contributions and insightful findings.**

A1: We respectfully disagree with the limited novelty assessment. **Our contribution is fundamentally data-centric rather than algorithmic**, introducing two key innovations: (1) A cross-domain mid-training framework that leverages data-rich non-GUI domains to enhance foundational abilities for data-scarce GUI tasks, and (2) As noted by Reviewers qdBH, yoKR, and L7Ty, novel empirical insights into which domain-specific data effectively enhances GUI agents' fundamental capabilities. These insights enable the community to approach GUI agent training data construction from a new perspective.

---

> ### Author Response · Authors · 2025-06-03
> **General Response - Contribution Declaration and Additional Benchmarks (2/2)**
>
> **Q2: Evaluation Strategy Rationale and Additional Benchmarks**
>
> A2: We clarify our benchmark selection rationale and provide additional experimental results:
>
> **(1) Benchmark Selection Rationale:**
>
>
> Our mid-training approach aims to enhance fundamental capabilities like knowledge and reasoning. Following OS-Genesis [1], We evaluate web and mobile environments as they are representative. We  **prioritize interactive benchmarks**  (AndroidWorld [2], WebArena [3])  **over static ones** (AndroidControl [4], AITW [5])  to better assess these fundamental capabilities rather than memorization of the annotators’ bias. Considering the time and resource limitations for action space alignment, we provide two additional real-world web benchmarks (Mind2Web-Live [6] and WebVoyager [7]) to evaluate the real-world abilities.
>
> **(2) Additional Experimental Results:** We conducted experiments on interactive, real-world Mind2Web-Live [6] and WebVoyager [7]. These evaluation results below are directly obtained with our previously trained models, without retraining specifically for these benchmarks.
>
> Mind2Web-Live: We evaluated 51 accessible tasks (remaining tasks excluded due to website access restrictions). Given label obsolescence from website updates, we use success rate as the primary metric.
>
> WebVoyager: We randomly sampled 10 tasks across 14 domains (140 total), excluding Google Search due to reCAPTCHA limitations. Human evaluation follows WebVoyager's standard protocol.
>
> | **Training Strategy** | **Success Rate** |
> |:----------------------:|:------------------:|
> | GUI Post-Training Only | 6.1 |
> | GUIMid + Post-Training | **10.2** |
>
> Table 1. Performance Comparison on Mind2Web-Live
>
> | **Model** | **Allrecipes** | **Amazon** | **Apple** | **ArXiv** | **BBC News** |
> |:-----------:|:----------------:|:------------:|:-----------:|:-----------:|:--------------:|
> | GUI Post-Training Only | 30.0 | 10.0 | 20.0 |  **50.0** | 10.0 |
> | GUIMid + Post-Training |  **50.0** |  **40.0** |  **40.0** | **50.0** |  **20.0** |
>
> | **Model** | **Booking** | **Cambridge Dictionary** | **Coursera** | **ESPN** | **GitHub** |
> |:-----------:|:----------------:|:------------:|:-----------:|:-----------:|:--------------:|
> | GUI Post-Training Only | 10.0 | 20.0 |  **50.0** | 10.0 | 20.0 |
> | GUIMid + Post-Training |  **20.0** |  **70.0** |  **50.0** |  **20.0** |  **40.0** |
>
> | **Model** | **Google Flights** | **Google Map** | **Huggingface** | **Wolfram Alpha** | **Average** |
> |:-----------:|:----------------:|:------------:|:-----------:|:-----------:|:--------------:|
> | GUI Post-Training Only | 0.0 | 10.0 | 30.0 | 20.0 | 20.7 |
> | GUIMid + Post-Training |  **10.0** | **20.0** |  **50.0** |  **50.0** |  **37.9** |
>
> Table 2. Performance Comparison on WebVoyager
>
> From Tables 1 and 2, our method demonstrates significant improvements in interactive real-world web environments. On Mind2Web-Live, our approach achieves a 67% relative improvement (10.2 vs 6.1). On WebVoyager, we observe an 83% relative improvement (37.9 vs 20.7) on average, with particularly strong gains in domains requiring complex reasoning such as Cambridge Dictionary (+250%) and Wolfram Alpha (+150%). These results indicate that our mid-training approach enhances fundamental capabilities that transfer effectively to diverse, dynamic web environments. We will include these experiments in the revised version of our paper.
>
>
> [1] Sun et al. OS-Genesis: Automating GUI Agent Trajectory Construction via Reverse Task Synthesis. ACL 2025 main.
> [2] Rawles, Christopher, et al. "Androidworld: A dynamic benchmarking environment for autonomous agents." arXiv preprint arXiv:2405.14573 (2024)
> [3] Zhou, Shuyan, et al. "WebArena: A Realistic Web Environment for Building Autonomous Agents." The Twelfth International Conference on Learning Representations.
> [4] Li, Wei, et al. "On the effects of data scale on ui control agents." Advances in Neural Information Processing Systems 37 (2024): 92130-92154.
> [5] Rawles, Christopher, et al. "Android The Wild: A large-scale dataset for android device control." Advances in Neural Information Processing Systems 36 (2023): 59708-59728.
> [6] Pan, Yichen, et al. "WebCanvas: Benchmarking Web Agents in Online Environments." Agentic Markets Workshop at ICML 2024.
> [7] He, Hongliang, et al. "WebVoyager: Building an End-to-End Web Agent with Large Multimodal Models." Proceedings of the 62nd Annual Meeting of the Association for Computational Linguistics (Volume 1: Long Papers). 2024.

---

### Decision · Program_Chairs · 2025-07-08

**Decision:**

Accept

**Comment:**

This work investigates how introducing a dedicated mid-training stage for Vision Language Models (VLMs) on diverse, reasoning-intensive non-GUI tasks can improve performance on GUI agent benchmarks. Reviewers acknowledged the significance of addressing data scarcity in GUI agent training by leveraging abundant external datasets spanning multimodal reasoning, textual reasoning, and GUI perception. The work provides novel empirical insights showing that cross-modal generalization occurs effectively—for example, text-only mathematical reasoning data significantly boosts visual GUI task performance. Extensive experiments across multiple benchmarks, including WebArena, AndroidWorld, Mind2Web-Live, and WebVoyager, demonstrated substantial performance gains of 8–12% absolute, with notable improvements in complex reasoning domains. However, some reviewers noted limited technical novelty, viewing the approach primarily as a data-centric empirical study rather than a methodological innovation. The authors responded with clarifications on their contribution as a data-centric approach, justification of benchmark choices consistent with prior work, additional experiments on interactive real-world web environments, and analysis of task-level improvements. Generally, reviewers appreciated the thorough empirical validation and actionable insights for future GUI agent training data design. I think this paper is a valuable contribution with strong empirical results and novel observations on cross-domain transfer, meriting acceptance despite some methodological conservatism.